

# Global scale variability of the mineral dust longwave refractive index:

# a new dataset of in situ measurements for climate modelling and remote sensing

Claudia Di Biagio[1], Paola Formenti[1], Yves Balkanski[2], Lorenzo Caponi[1,3], Mathieu Cazaunau[1], Edouard Pangui[1], Emilie Journet[1], Sophie Nowak[4], Sandrine Caquineau[5], Meinrat O. Andreae[6,12], Konrad Kandler[7], Thuraya Saeed[8], Stuart Piketh[9], David Seibert[10], Earle Williams[11], and Jean-Francois Doussin[1]

[1] *Laboratoire Interuniversitaire des Systèmes Atmosphériques (LISA), UMR 7583, CNRS, Université Paris Est Créteil et Université Paris Diderot, Institut Pierre et Simon Laplace, Créteil, France*

[2] *Laboratoire des Sciences du Climat et de l'Environnement, CEA CNRS UVSQ, 91191, Gif sur Yvette, France*

[3] *University of Genoa, Genoa, Italy*

[4] *Plateforme RX UFR de chimie, Université Paris Diderot, Paris, France*

[5] *IRD-Sorbonne Universités (UPMC, Univ. Paris 06) – CNRS-MNHN, LOCEAN Laboratory, IRD France-Nord, F-93143 Bondy, France*

[6] *Biogeochemistry Department, Max Planck Institute for Chemistry, P.O. Box 3060, 55020, Mainz, Germany*

[7] *Institut für Angewandte Geowissenschaften, Technische Universität Darmstadt, Schnittspahnstr. 9, 64287 Darmstadt, Germany*

[8] *Science department, College of Basic Education, Public Authority for Applied Education and Training, Al-Ardeya, Kuwait*

[9] *Climatology Research Group, Unit for Environmental Science and Management, North-West University, Potchefstroom, South Africa*

[10] *Walden University, Minneapolis, Minnesota, USA*

[11] *Parsons Laboratory, Massachusetts Institute of Technology, Cambridge, Massachusetts, USA*

[12] *Department of Geosciences, King Saud University, Riyadh, Saudi Arabia*

Correspondence to:

C. Di Biagio (claudia.dibiagio@lisa.u-pec.fr) and P. Formenti (paola.formenti@lisa.u-pec.fr)





**Abstract**

Modelling the interaction of dust with longwave (LW) radiation is still a challenge due to the scarcity of information on the complex refractive index of dust from different source regions. In particular, little is known on the variability of the refractive index as a function of the dust mineralogical composition, depending on the source region of emission, and the dust size distribution, which is modified during transport. As a consequence, to date, climate models and remote sensing retrievals generally use a spatially-invariant and time-constant value for the dust LW refractive index.

In this paper the variability of the mineral dust LW refractive index as a function of its mineralogical composition and size distribution is explored by in situ measurements in a large smog chamber. Mineral dust aerosols were generated from nineteen natural soils from Northern Africa, Sahel, Middle East, Eastern Asia, North and South America, Southern Africa, and Australia. Soil samples were selected from a total of 137 samples available in order to represent the diversity of sources from arid and semi-arid areas worldwide and to account for the heterogeneity of the soil composition at the global scale. Aerosol samples generated from soils were re-suspended in the chamber, where their LW extinction spectra (2-16 µm), size distribution, and mineralogical composition were measured. The generated aerosol exhibits a realistic size distribution and mineralogy, including both the sub- and super-micron fractions, and represents in typical atmospheric proportions the main LW-active minerals, such as clays, quartz, and calcite. The complex refractive index of the aerosol is obtained by an optical inversion based upon the measured extinction spectrum and size distribution.

Results from the present study show that the LW refractive index of dust varies greatly both in magnitude and spectral shape from sample to sample, following the changes in the measured particle composition. The real part (n) of the refractive index is between 0.84 and 1.94, while the imaginary part (k) is ~0.001 and 0.92. For instance, the strength of the absorption at ~7 and 11.4 µm depends on the amount of calcite within the samples, while the absorption between 8 and 14 µm is determined by the relative abundance of quartz and clays. A linear relationship between the magnitude of the refractive index at 7.0, 9.2, and 11.4 µm and the mass concentration of calcite and quartz absorbing at these wavelengths was found. We suggest that this may lead to predictive rules to estimate the LW refractive index of dust in specific bands based on an assumed or predicted mineralogical composition, or conversely, to estimate the dust composition from measurements of the LW extinction at specific wavebands.





Based on the results of the present study, we recommend using refractive indices specific for the different source regions, rather than generic values, in climate models and remote sensing applications. Our observations also suggest that the refractive index of dust in the LW does not change due to the loss of coarse particles by gravitational settling, so that a constant value could be assumed close to sources and during transport. The results of the present study also clearly suggest that the LW refractive index of dust varies at the regional scale. This regional variability has to be characterized further in order to better assess the influence of dust on regional climate, as well as to increase the accuracy of satellite retrievals over regions affected by dust.

We make the whole dataset of the dust complex refractive indices obtained here available to the scientific community by publishing it in the supplementary material to this paper.

**Keywords**: mineral dust, longwave refractive index, mineralogy, size distribution, global variability

## 1. Introduction

Mineral dust is one of the most abundant aerosol species in the atmosphere and contributes significantly to radiative perturbation, both at the regional and the global scale (Miller et al., 2014). The direct radiative effect of mineral dust acts both at shortwave (SW) and longwave (LW) wavelengths (Tegen and Lacis, 1996). This is due to the very large size spectrum of these particles, which extends from hundreds of nanometers to tenths of micrometers, and to their mineralogy, which includes minerals with absorption bands at both SW and LW wavelengths (Sokolik et al., 1998; Sokolik and Toon, 1999). The sub-micron dust fraction controls the interaction in the SW, where scattering is the dominant process, while the super-micron size fraction drives the LW interaction, dominated by absorption (Sokolik and Toon, 1996 and 1999). The SW and LW terms have opposite effects at the surface, Top-of-Atmosphere (TOA), and within the atmosphere (Hsu et al., 2000). Indeed, the dust SW effect is to cool the surface and the TOA, and to warm the atmosphere; conversely, the dust LW effect induces a warming of the surface and TOA, and an atmospheric cooling.

The interaction of dust with LW radiation has key implications for climate modelling and remote sensing. Many studies have shown the key role of the LW effect in modulating the SW perturbation of dust



not only close to sources (Slingo et al., 2006), where the coarse size fraction is dominant (Schütz et al,
1974; Ryder et al., 2013a), but also after medium- and long- range transport (di Sarra et al., 2011;
Meloni et al., 2015), when the larger particles (> 10 μm) were preferentially removed by wet and dry
deposition (Schütz et al, 1981; Maring et al., 2003; Osada et al., 2014). Thus, the dust LW term has
importance over the entire dust lifecycle, and has to be taken into account in order to evaluate the radi-
ative effect of dust particles on the climate system. Second, the signature of the dust LW absorption
modifies the TOA radiance spectrum, which influences the retrieval of several climate parameters by
satellite remote sensing. Misinterpretations of the data may occur if the signal of dust is not accurately
taken into account within satellite inversion algorithms (Sokolik, 2002; DeSouza-Machado et al.,
2006; Maddy et al., 2012). In addition, the dust LW signature obtained by spaceborne satellite data in
the 8–12 μm window region is used to estimate the concentration fields and optical depth of dust (Klü-
ser et al., 2011; Capelle et al., 2014; Cuesta et al., 2015), with potential important applications for cli-
mate and air quality studies, health issues, and visibility.
Currently, the magnitude and the spectral fingerprints of the dust signal in the LW are still very uncer-
tain. The highest uncertainty comes from the poor knowledge on the dust spectral complex refractive
index (m= n-ik) (Claquin et al., 1998; Liao and Seinfeld, 1998; Sokolik et al., 1998; Highwood et al.,
2003; Colarco et al., 2014). The dust complex refractive index in the LW depends on the particle min-
eralogical composition, in particular the relative proportion of quartz, clays (kaolinite, illite, smectite,
chlorite), and calcium-rich minerals (calcite, dolomite), each exhibiting specific absorption features in
the LW spectrum (Sokolik et al., 1993 and 1998). Because of the variability of the dust composition
resulting from the variability of composition of the source soils (Jeong, 2008; Scheuvens et al., 2013;
Formenti et al., 2014; Journet et al., 2014), atmospheric dust produced from different regions of the
world is expected to have a varying complex refractive index. Additional variability is expected to be
introduced during transport due to the progressive loss of coarse particles by gravitational settling and
processing, which both change the mineralogical composition (Pye et al., 1987; Usher et al., 2003). As
a consequence, the refractive index of dust is expected to vary widely at the regional and global scale.
Several studies have recommended taking into account the variability of the dust LW refractive index
in order to correctly represent its effect in climate models and satellite retrieval algorithms (Sokolik et
al., 1998; Claquin et al., 1999; Balkanski et al., 2007; Colarco et al., 2014; Capelle et al., 2014; among
others). However, to date this is precluded by the limited body of observations available. Most past
studies on the LW refractive index have been performed on single synthetic minerals (see Table 1 in





Otto et al., 2009). These data, however, are not adequate to reproduce atmospheric dust because of the
chemical differences between the reference minerals and the minerals in the natural aerosol, and also
because of the difficulty of effectively evaluating the refractive index of the dust aerosol based only on
information on its single constituents (e.g., McConnell et al., 2010). On the other hand, very few stud-
ies have been performed on natural aerosol samples. They include the estimates obtained with the KBr
pellet technique by Volz (1972, 1973), Fouquart (1987), and, more recently, by Di Biagio et al.
(2014a), on dust samples collected at a few geographical locations (Germany, Barbados, Niger, and
Algeria). Besides hardly representing global dust sources, these datasets are also difficult to extrapo-
late to atmospheric conditions as (i) they mostly refer to unknown dust mineralogical composition and
size distribution, and also (ii) are obtained from analyses of field samples that might have experienced
unknown physico-chemical transformations. In addition, they have a rather coarse spectral resolution,
which is sometimes insufficient to resolve the main dust spectral features.
As a consequence, climate models and satellite retrievals presently use a spatially-invariant and time-
constant value for the dust LW refractive index (e.g., Miller et al., 2014; Capelle et al., 2014), implicit-
ly assuming a uniform as well as  transport- and processing-invariant dust composition.
Recently, novel data of the LW refractive index for dust from the Sahara, the Sahel, and the Gobi de-
serts have been obtained from in situ measurements in a large smog chamber (Di Biagio et al., 2014b;
hereinafter DB14). These measurements were performed in the realistic and dynamic environment of
the 4.2 m$^3$ CESAM chamber (French acronym for Experimental Multiphasic Atmospheric Simulation
Chamber) (Wang et al., 2011), using a validated generation mechanism to produce mineral dust from
parent soils (Alfaro et al., 2004). The mineralogical composition and size distribution of the particles
were measured along with the optical data, thus providing a link between particle physico-chemical
and optical properties.
In this study, we review, optimize, and extend the approach of DB14 to investigate the LW optical
properties of mineral dust aerosols from nineteen soils from major source regions worldwide, in order
to: (i) characterize the dependence of the dust LW refractive index on the particle origin and different
mineralogical compositions; and (ii) investigate the variability of the refractive index as a function of
the change in size distribution that may occur during medium- and long-range transport.
The paper is organized as follows: in Sect. 2 we describe the experimental set-up, instrumentation and
data analysis, while in Sect. 3 the algorithm to retrieve the LW complex refractive index from observa-



tions is discussed. Criteria for soil selection and their representativeness of the global dust are dis-
cussed in Sect. 4. Results are presented in Sect. 5. At first, the atmospheric representativeness in terms
of mineralogy and size distribution of the generated aerosols used in the experiments is evaluated
(Sect. 5.1 and 5.2), then the extinction and complex refractive index spectra obtained for the different
source regions and at different aging times in the chamber are presented in Sect. 5.3. The discussion of
the results, their comparison with the literature, and the main conclusions are given in Sect. 6 and 7.

## 2. Experimental set-up and instrumentation

The schematic configuration of the CESAM chamber set-up for the dust experiments is shown in Fig.
1. Prior to each experiment, the chamber was evacuated and kept at a pressure of $3 \cdot 10^{-4}$ hPa. Then, the
reactor was filled with a mixture of 80% $N_2$ (produced by evaporation from a pressurized liquid nitro-
gen tank, Messer, purity >99.995%) and 20% $O_2$ (Linde, 5.0). The chamber was equipped with a four-
blade stainless steel fan to achieve homogeneous conditions within the chamber volume (with a typical
mixing time of approximately 1 minute). Mineral dust aerosols generated from parent soils were dis-
persed into the chamber and left in suspension for a time period of 60-120 min, whilst monitoring the
evolution of their physico-chemical and optical properties. The LW spectrum of the dust aerosols was
measured by means of an in situ FTIR. Concurrently, the particle size distribution and the SW scatter-
ing and absorption coefficients were measured by several instruments sampling aerosols from the
chamber. They include a scanning mobility particle sizer (SMPS), and WELAS and SkyGrimm optical
particle counters for the size distribution, and a nephelometer (TSI Inc. model 3563), an aethalometer
(Magee Sci. model AE31), and two Cavity Attenuated Phase Shift Extinction (CAPS PMeX by Aero-
dyne) for aerosol SW optical properties. Dust samples were also collected over the largest part of each
experiment on polycarbonate filters (47-mm Nuclepore, Whatman, nominal pore size 0.4 μm) for an
analysis of the particle mineralogical composition averaged over the length of the experiment.
Inlets for the instruments sampling aerosols from the chamber (size, SW optics, filter sampling) con-
sisted of two parts: 1) a stainless steel tube (~20-40 cm length, 9.5 mm diameter) located inside
CESAM which extracted air from the interior of the chamber and 2) an external connection from the
chamber to the instruments. All external connections were made using 0.64 cm conductive silicone
tubing (TSI Inc.) that minimizes particle loss by electrostatic deposition. The sampling lines were de-
signed to be as straight and as short as possible, and their total length varied between 40 and 120 cm.



The possible losses as a function of particle diameter were carefully estimated for each inlet and the
related data properly corrected (Sect. 2.3.2). To compensate for the air being extracted from the cham-
ber by the various instruments, a particle-free $N_2/O_2$ mixture was continuously injected into the cham-
ber.
All experiments were conducted at ambient temperature and relative humidity <2%. The chamber was
manually cleaned between the different experiments to avoid any carryover contaminations as far as
possible. Background concentrations of aerosols in the chamber varied between 0.5 and 2.0 µg m$^{-3}$.
In the following paragraphs we describe the system for dust generation, measurements of dust LW
spectrum, size distribution, and mineralogy, and data analysis. A summary of the different measured
and retrieved quantities in this study and their estimated uncertainties is reported in Table 1. Longwave
optical and size distribution data, acquired with different temporal resolutions, are averaged over 10-
min intervals. Uncertainties on the average values are obtained as the standard deviation over the 10-
min intervals.
A full description of the SW optical measurements and results is out of the scope of the present study
and will be provided in a forthcoming paper (Di Biagio et al., in preparation).

## 198   2.1 Dust aerosol generation

In order to mimic the natural emission process, dust aerosols were generated by mechanical shaking of
natural soil samples as described in DB14. The soils used in this study consist of the surface layer,
which is subject to wind erosion in nature (Pye et al., 1987). Prior to each experiment, the soil samples
were sieved to <1000 µm and dried at 100 °C for about 1 h to remove any residual humidity. This pro-
cessing did not affect the mineral crystalline structure of the soil (Sertsu and Sánchez, 1978).
About 15 g of soil sample was placed in a Büchner flask and shaken for about 30 min at 100 Hz by
means of a sieve shaker (Retsch AS200). The dust suspension in the flask was then injected into the
chamber by flushing it with $N_2$ at 10 L min$^{-1}$ for about 10-15 min, whilst continuously shaking the soil.
Larger quantities of soil sample (60 g) mixed with pure quartz (60 g) had been used in DB14 to max-
imize the concentrations of the generated dust. The presence of the pure quartz grains increases the
efficiency of the shaking, allowing a rapid generation of high dust concentrations. In that case it had
been necessary, however, to pass the aerosol flow through a stainless steel settling cylinder to avoid





large quartz grains from entering the chamber (DB14). For the present experiments the generation sys-
tem was optimized, i.e. the mechanical system used to fix the flask to the shaker was improved so that
the soil shaking was more powerful, and sufficient quantities of dust aerosols could be generated by
using a smaller amount of soil and without adding quartz to the soil sample. In this way, the settling
cylinder could be eliminated. No differences were observed in the size distribution or mineralogy of
the generated dust between the two approaches.

**2.2 LW optical measurements: FTIR extinction spectrum**
The extinction spectrum of dust aerosols in the longwave was measured by means of an in situ Fourier
Transform Infrared spectrometer (FTIR) (Bruker® Tensor 37[TM]) analytical system. The spectrometer
is equipped with a liquid nitrogen-cooled Mercury Cadmium Telluride (MCT) detector and a Globar
source. The FTIR measures between wavelengths of 2.0 µm (5000 cm$^{-1}$) and 16 µm (625 cm$^{-1}$) at 2
cm$^{-1}$ resolution by co-adding 158 scans over 2 minutes. The FTIR is interfaced with a multi-pass cell
to achieve a total optical path length ($x$) within the chamber of 192 ± 4 m. The FTIR reference spec-
trum was acquired immediately before the dust injection. In some cases small amounts of water vapor
and $CO_2$ entered CESAM during particle injection and partly contaminated the dust spectra below 7
µm. This did not influence the state of particles as the chamber remained very dry (relative humidity <
2%). Water vapor and $CO_2$ absorption lines were carefully subtracted using reference spectra. The
measured spectra were then interpolated at 0.02 µm wavelength resolution. Starting from the FTIR
measured transmission (T), the dust spectral extinction coefficient $\beta_{ext}$ in the 2-16 µm range was calcu-
lated as:
$$\beta_{ext}(\lambda) = \frac{-\ln\left(T(\lambda)\right)}{x}. \quad (1)$$

The uncertainty on $\beta_{ext}$ was calculated with the error propagation formula by considering the uncertain-
ties arising from T noise (~1%) and from the standard deviation of the 10-min averages and of the path
length $x$. We estimated it to be ~10%.
In the 2-16 µm range the dust extinction measured by the FTIR is due to the sum of scattering and ab-
sorption. Scattering dominates below 6 µm, while absorption is dominant above 6 µm. The FTIR mul-
tipass cell in the CESAM chamber has been built following the White (1942) design (see Fig. 1). In
this configuration, a significant fraction of the light scattered by the dust enters the FTIR detector and



is not measured as extinction. This is because mineral dust is dominated by the super-micron fraction,
which scatters predominantly in the forward direction. As a consequence, the FTIR signal in the pres-
ence of mineral dust will represent only a fraction of dust scattering below 6 μm and almost exclusive-
ly absorption above 6 μm. Figure S1 (supplementary material), shows an example of the angular dis-
tribution of scattered light (phase function) and the scattering-to-absorption ratio calculated as a func-
tion of the wavelength in the LW for one of the samples used in this study. Results of the calculations
confirm that above 6 μm the scattering signal measured by the FTIR accounts for less than 20% of the
total LW extinction at the peak of the injection and less than 10% after 120 minutes in the chamber.
Consequently, we approximate Eq. (1) as:
$$\beta_{abs}(\lambda) \approx \frac{-\ln(T(\lambda))}{x} \quad (\lambda > 6 \ \mu m). \qquad (2)$$


### 251 2.3 Size distribution measurements

The particle number size distribution in the chamber was measured with several instruments based on
different principles and operating in different size ranges:
- a scanning mobility particle sizer (SMPS) (TSI, DMA Model 3080, CPC Model 3772; operated at
2.0/0.2 L min$^{-1}$ sheath/aerosol flow rates; 2-min resolution) measuring the dust electrical mobili-
ty diameters ($D_m$, i.e., the diameter of a sphere with the same migration velocity in a constant
electric field as the particle of interest) in the range 0.019–0.882 μm. Given that dust particles
have a density larger than unity (assuming an effective density of 2.5 g cm$^{-3}$), the cut point of the
impactor at the input of the SMPS shifts towards lower diameters. This reduces the range of
measured mobility diameters to ~0.019-0.50 μm. The SMPS was calibrated prior the campaign
with PSL particles (Thermo Sci.) of 0.05, 0.1, and 0.5 μm nominal diameters;
- a WELAS optical particle counter (PALAS, model 2000; white light source between 0.35-0.70 μm;
flow rate 2 L min$^{-1}$; 1-min resolution) measuring the dust sphere-equivalent optical diameters
($D_{opt}$, i.e., the diameter of a sphere yielding on the same detector geometry the same optical re-
sponse as the particle of interest) in the range 0.58-40.7 μm. The WELAS was calibrated prior
the campaign with Caldust 1100 (Palas) reference particles;
-  a SkyGrimm optical particle counter (Grimm Inc., model 1.129; 0.655 µm operating wavelength;

flow rate 1.2 L min$^{-1}$; 6-sec resolution) measuring the dust sphere-equivalent optical diameters

($D_{opt}$) in the range 0.25-32 µm. The SkyGrimm was calibrated after the campaign against a

"master" Grimm (model 1.109) just recalibrated at the factory.

The SMPS and the WELAS were installed at the bottom of the chamber, while the SkyGrimm was
installed at the top of the chamber on the same horizontal plane as the FTIR spectrometer and at about
60 cm across the chamber from the WELAS and the SMPS. As already discussed in DB14, measure-
ments at the top and bottom of the chamber are in very good agreement during the whole duration of
each experiment, which indicates a good homogeneity of the dust aerosols in the chamber.

### 2.3.1 Corrections of SMPS, WELAS, and SkyGrimm data

Different corrections have to be applied to the instruments measuring the particle size distribution. For
the SMPS, corrections for particle loss by diffusion in the instrument tubing and the contribution of
multiple-charged particles were performed using the SMPS software. The electrical mobility diameter
measured by the SMPS was converted to a geometrical diameter ($D_g$) by taking into account the parti-
cle dynamic shape factor ($\chi$), as $D_m = D_m / \chi$. The shape factor $\chi$, determined by comparison with the
SkyGrimm in the overlapping particle range (~0.25-0.50 µm), was found to be 1.75±0.10. This value
is higher than those reported in the literature for mineral dust (1.1-1.6; e.g., Davies, 1979; Kaaden et
al., 2008). The uncertainty in $D_g$ was estimated with the error propagation formula and was ~6%.
For the WELAS, optical diameters were converted to sphere-equivalent geometrical diameters ($D_g$) by
taking into account the visible complex refractive index. The $D_{opt}$ to $D_g$ diameter conversion was per-
formed based on the range of values reported in the literature for dust in the visible range, i.e., 1.47–
1.53 for the real part and 0.001–0.005 for the imaginary part (Osborne et al., 2008; Otto et al., 2009;
McConnell et al., 2010; Kim et al., 2011; Klaver et al., 2011). Optical calculations were computed
over the spectral range of the WELAS using Mie theory for spherical particles by fixing n at 1.47, 1.50
and 1.53 and by varying k in steps of 0.001 between 0.001 and 0.005. The spectrum of the WELAS
lamp needed for optical calculations was measured in the laboratory (Fig. S2, supplementary material).
$D_g$ was then set at the mean ± one standard deviation of the values obtained for the different n and k.
After calculations, the WELAS $D_g$ range became 0.65-73.0 µm with an associated uncertainty of <5%



for $D_g$<10 μm and between 5 and 7% at larger diameters. A very low counting efficiency was observed
for the WELAS below 1 μm, thus data in this size range were discarded.
For the SkyGrimm, the $D_{opt}$ to $D_g$ diameter conversion was performed with a procedure similar to that
used for the WELAS. After calculations, the $D_g$ range for the SkyGrimm became 0.29-68.2 μm with
an associated uncertainty <15.2% at all diameters. The inter-calibration between the SkyGrimm and
the master instrument showed a relatively good agreement (<20% difference in particle number) at
$D_g$<1 μm, but a large disagreement (up to 300% difference) at $D_g$>1 μm. Based on inter-comparison
data, a recalibration curve was calculated for the SkyGrimm in the range $D_g$<1 μm, and the data for
$D_g$>1 μm were discarded. The SkyGrimm particle concentration was also corrected for the flow rate of
the instrument, which during the experiment was observed to vary between 0.7 and 1.2 L min$^{-1}$ com-
pared to its nominal value at 1.2 L min$^{-1}$.

**2.3.2 Correction for particle losses in sampling lines and determination of the full dust size dis-**
**tribution at the input of each instrument**
In order to compare and combine extractive measurements (size distribution, filter sampling, and SW
optics), particle losses due to aspiration and transmission in the sampling lines were calculated using
the Particle Loss Calculator (PLC) software (von der Weiden et al., 2009). Inputs to the software in-
clude the geometry of the sampling line, the sampling flow rate, the particle shape factor χ, and the
particle density (set at 2.5 g cm$^{-3}$ for dust).
Particle losses for the instruments measuring the number size distribution (SMPS, WELAS, and
SkyGrimm) were calculated. This allowed reconstructing the dust size distribution suspended in the
CESAM chamber that corresponds to the size distribution sensed by the FTIR and that is needed for
optical calculations in the LW. Particle loss was found negligible at $D_g$<1 μm, reaching 50% at $D_g$~5
μm, 75% at $D_g$ ~6.3 μm, and 95% at $D_g$ ~8 μm for the WELAS, the only instrument considered in the
super-micron range. Data for the WELAS were then corrected as
$$\left[ dN / d\log D_g \right]_{Corr,WELAS} = \left[ dN / d\log D_g \right]_{WELAS} / \left[ 1 - L_{WELAS}\left( D_g \right) \right] \quad (3)$$

where [dN/dlogD$_g$]$_{WELAS}$ is the size measured by the WELAS and $L_{WELAS}(D_g)$ is the calculated particle
loss as a function of the particle diameter. Data at $D_g$>8 μm, for which the loss is higher than 95%,
were excluded from the dataset due to their large uncertainty. The uncertainty on $L_{WELAS}(D_g)$ was es-





timated with a sensitivity study by varying the PLC software values of the input parameters within
their error bars. The $L_{WELAS}(D_g)$ uncertainty varies between ~50% at 2 μm to ~10% at 8 μm. The total
uncertainty in the WELAS-corrected size distribution was estimated as the combination of the
$dN/dlogD_g$ standard deviation on the 10-min average and the $L_{WELAS}(D_g)$ uncertainty.
The full size distribution of dust aerosols within the CESAM chamber $\left[dN/d\log D_g\right]_{CESAM}$ was deter-
mined by combining SMPS and SkyGrimm data with WELAS loss-corrected data: the SMPS was tak-
en at $D_g<0.3$ μm, the SkyGrimm at $D_g=0.3-1.0$ μm, and the WELAS at $D_g=1.0-8.0$ μm. Data were then
interpolated in steps of $dlogD_g=0.05$. An example of the size distributions measured by the different
instruments is shown in Fig. S3 in the supplement for this paper. Above 8 μm, where WELAS data
were not available, the dust size distribution was extrapolated by applying a single-mode lognormal fit.
The fit was set to reproduce the shape of the WELAS distribution between $D_g$~3-4 and 8 μm.
Particle losses in the filter sampling system ($L_{filter}(D_g)$) were calculated estimating the size-dependent
particles losses that would be experienced by an aerosol with the size distribution in CESAM recon-
structed from the previous calculations. Losses for the sampling filter were negligible for $D_g<1$ μm,
and increased to 50% at $D_g$ ~6.5 μm, 75% at $D_g$ ~9 μm, and 95% at $D_g$ ~12 μm. The loss function,
$L_{filter}(D_g)$, was used to estimate the dust size distribution at the input of the filter sampling system as
$$\left[dN/d\log D_g\right]_{filter} = \left[dN/d\log D_g\right]_{CESAM} * \left[1-L_{filter}\left(D_g\right)\right] \quad (4).$$

As a consequence of losses, the FTIR and the filters sense particles over different size ranges. Figure
S4 (supplementary material) illustrates this point by showing a comparison of the calculated size dis-
tribution within CESAM and that sampled on filters for one typical case. An underestimation of the
particle number on the sampling filter compared to that measured in CESAM is observed above 10 μm
diameter. While the filter samples would underestimate the mass concentration in the chamber, the
relative proportions of the main minerals should be well represented. As a matter of fact, at emission,
where particles of diameters above 10 μm are most relevant, the mineralogical composition in the 10-
20 μm size class matches that of particles of diameters between 5 and 10 μm (Kandler et al., 2009).
When averaging, and also taking into account the contribution of the mass of the 10-20 μm size class
to the total, differences in the relative proportions of minerals do not exceed 10%.




### 2.4 Analysis of the dust aerosol mineralogical composition

The mineralogical composition of the aerosol particles collected on the filters was determined by X-Ray Diffraction (XRD) analysis. XRD analysis was performed using a Panalytical model Empyrean diffractometer with Ni-filtered CuK$_\alpha$ radiation at 45 kV and 40 mA. Samples were scanned from 5 to 60° (2θ) in steps of 0.026°, with a time per step of 200 s. Samples were prepared and analyzed according to the protocols of Caquineau et al. (1997) for low mass loadings (load deposited on filter <800 μg). Particles were first extracted from the filter with ethanol, then concentrated by centrifuging (25,000 rpm for 30 min), diluted with deionized water (pH ~ 7.1), and finally deposited on a pure silicon slide.

For well-crystallized minerals, such as quartz, calcite, dolomite, gypsum and feldspars (orthoclase, albite), a mass calibration was performed in order to establish the relationship between the intensity of the diffraction peak and the mass concentration in the aerosol samples, according to the procedure described in Klaver et al. (2011). The calibration coefficients $K_i$, representing the ratio between the total peak surface area in the diffraction spectra ($S_i$) and the mass $m_i$ of the $i^{th}$-mineral, are reported in Table S1 in the supplementary material. The error in the obtained mass of each mineral was estimated with the error propagation formula taking into account the uncertainty in $S_i$ and the calibration coefficients $K_i$. The obtained uncertainty is ±9% for quartz, ±14% for orthoclase, ±8% for albite, ±11% for calcite, ±10% for dolomite, and ±18% for gypsum.

Conversely, the mass concentration of clays (kaolinite, illite, smectite, palygorskite, chlorite), also detected in the samples, cannot be quantified in absolute terms from the XRD spectra due to the absence of appropriate calibration standards for these components (Formenti et al., 2014). Hence, the total clay mass was estimated as the difference between the total dust mass calculated from particle size distribution $\left[ dN / d \log D_g \right]_{filter}$ and the total mass of quartz, calcium-rich species, and feldspars estimated after calibration. The mass of organic material was neglected, as well as that of iron and titanium oxides, whose contributions should not exceed 5% according to literature (Lepple and Brine, 1976; Lafon et al., 2006; Formenti et al., 2014). The uncertainty on the estimated total clay mass, estimated between 8 and 26%, was calculated with the error propagation formula including the uncertainties on total dust mass and on the mass of each identified mineral.

For the Northern African and Eastern Asian aerosols, the mass apportionment between the different clay species was based on literature values of illite-to-kaolinite (I/K) and chlorite-to-kaolinite (Ch/I)



mass ratios (Scheuvens et al., 2013; Formenti et al. 2014). For the other samples, only the total clay
mass was estimated.

### 3. Retrieval of the LW complex refractive index

An optical inversion procedure was applied to retrieve the LW complex refractive index (m=n–ik) of
the dust aerosols based on the simultaneous measurements of the particle LW spectra and size. Starting
from the number size distribution, $\left[ dN / d\log D_g \right]_{CESAM}$, the LW absorption coefficient, $\beta_{abs}(\lambda)$, meas-
ured in CESAM can be calculated as:
$$\left(\beta_{abs}(\lambda)\right)_{calc} = \sum_{D_g} \frac{\pi D_g^2}{4} Q_{abs}(m,\lambda,D_g)\left[\frac{dN}{d\log D_g}\right]_{CESAM} d\log D_g \qquad (5)$$

where $Q_{abs}(m,\lambda,D_g)$ is the particle absorption efficiency. As the simplest approach, $Q_{abs}$ can be com-
puted using Mie theory for spherical particles.
Our retrieval algorithm consists of iteratively varying m in expression (5) until $(\beta_{abs}(\lambda))_{calc}$ matches the
measured $\beta_{abs}(\lambda)$. However, as m is a complex number with two variables, an additional condition is
needed. According to electromagnetic theory, n and k must satisfy the Kramers-Kronig (K-K) relation-
ship (Bohren and Huffmann, 1983):
$$n(\omega)-1 = \frac{2}{\pi} P \int_0^\infty \frac{\Omega \cdot k(\Omega)}{\Omega^2 - \omega^2} \cdot d\Omega \quad (6)$$

with $\omega$ the angular frequency of radiation ($\omega=2\pi c/\lambda$, [s$^{-1}$]), and P the principal value of the Cauchy in-
tegral. Equation (6) means that if $k(\lambda)$ is known, then $n(\lambda)$ can be calculated accordingly. Hence, the K-
K relation is the additional condition beside (5) to retrieve n and k. A direct calculation of the K-K
integral is, however, very difficult as it requires the knowledge of k over an infinite wavelength range.
A useful formulation, which permits one to obtain the couple of n-k values that automatically satisfy
the K-K condition, is the one based on the Lorentz dispersion theory. In the Lorentz formulation, n and
k may be written as a function of the real ($\varepsilon_r$) and imaginary ($\varepsilon_i$) parts of the particle dielectric function
as:
$$n(\omega) = \left(\frac{1}{2}\left[\sqrt{\left(\varepsilon_r(\omega)\right)^2 + \left(\varepsilon_i(\omega)\right)^2} + \varepsilon_r(\omega)\right]\right)^{1/2} \quad (7a)$$



$$k(\omega) = \left( \frac{1}{2} \left[ \sqrt{(\varepsilon_r(\omega))^2 + (\varepsilon_i(\omega))^2} - \varepsilon_r(\omega) \right] \right)^{1/2} \quad (7b)$$

$\varepsilon_r(\omega)$ and $\varepsilon_i(\omega)$ can be in turn expressed as the sum of N Lorentzian harmonic oscillators:
$$\varepsilon_r(\omega) = \varepsilon_\infty + \left[ \sum_{j=1}^{N} \frac{F_j(\omega_j^2 - \omega^2)}{(\omega_j^2 - \omega^2)^2 + \gamma_j^2 \omega^2} \right] \quad (8a)$$

$$\varepsilon_i(\omega) = \sum_{j=1}^{N} \frac{F_j \gamma_j \omega}{(\omega_j^2 - \omega^2)^2 + \gamma_j^2 \omega^2} \quad (8b)$$

where $\varepsilon_\infty = n_{vis}^2$ is the real dielectric function in the limit of visible wavelengths, and $n_{vis}$ the real part
of the refractive index in the visible, and $(\omega_j, \gamma_j, F_j)$ are the three parameters (eigenfrequency, damping
factor, and strength) characterizing the j-th oscillator.
In our algorithm we combined (7a)-(7) and (8a)-(8b) with (5) to retrieve n-k values that allow both to
reproduce the measured $\beta_{abs}(\lambda)$ and to satisfy the K-K relationship. In practice, in the iteration proce-
dure only one of the two components of the refractive index (in our case, k) was varied, while the other
(n) was recalculated at each step based on the values of the oscillator parameters $(\omega_j, \gamma_j, F_j)$ obtained
from a best fit for k. In the calculations, the initial value of $k(\lambda)$ was set at $k(\lambda) = \lambda \beta_{abs}(\lambda)/4\pi$, then in the
iteration procedure, $k(\lambda)$ was varied in steps of 0.001 without imposing any constraint on its spectral
shape. Initial values of the $(\omega_j, \gamma_j, F_j)$ parameters were manually set based on the initial spectrum of
$k(\lambda)$. Between 6 and 10 oscillators were needed to model the $k(\lambda)$ spectrum for the different cases. The
fit between $k(\lambda)$ and Eq. (7b) was performed using the Levenberg-Marquardt technique. The iteration
procedure was stopped when the condition: $|(\beta_{abs}(\lambda))_{calc} - \beta_{abs}(\lambda)| < 1\%$ is met at all wavelengths.
Optical calculations were performed between 6 and 16 μm, within a range where FTIR measured scat-
tering could be neglected (see Sect. 2.2). Below 6 μm, $k(\lambda)$ was then fixed to the value obtained at 6
μm. Calculations were performed over 10-min intervals.
For each experiment and for each 10-min interval, the value of $n_{vis}$ to use in Eq. (8a) was obtained
from optical calculations using the simultaneous measurements of the SW scattering and absorption
coefficients performed in CESAM (Di Biagio et al., in preparation). For the various aerosol samples
considered here the value of $n_{vis}$ varied between 1.47 and 1.52 with an uncertainty <2%. This approach
is better than the one used in DB14, where the value of $n_{vis}$ was manually adjusted for successive trials.





Specifically, in DB14, $n_{vis}$ was varied and set to the value that allowed best reproducing the measured
dust scattering signal below 6 μm. As discussed in Sect. 2.2, however, only a fraction of the total dust
scattering is measured by the FTIR. As a result, the $n_{vis}$ values obtained in DB14 were considerably
lower that the values generally assumed for dust ($n_{vis}$ =1.32-1.35 compared to 1.47-1.53 from the liter-
ature, e.g., Osborne et al., 2008; McConnell et al., 2010), with a possible resulting overall underestima-
tion of n. Here, instead, the $n_{vis}$ value was obtained based on additional SW optical measurements,
which ensured a more reliable estimate of the whole spectral n.
The validity of the proposed retrieval procedure was assessed by performing a control experiment
where ammonium sulfate aerosols were injected in the chamber. Ammonium sulfate has been widely
studied in the past and its optical properties are well known (e.g., Toon et al., 1976; Flores et al.,
2009). The description and the results of the control experiment are reported in Appendix 1.

**3.1 Caveats on the retrieval procedure for the LW refractive index**
The procedure for the retrieval of the complex refractive index presented in the previous section com-
bines optical calculations, the Kramers-Kronig relation and the Lorentz dispersion theory, and was
based on measurements of spectral absorption and particle size distribution. The approach is quite sen-
sitive to the accuracy and representativeness of the measurements and assumptions in the optical calcu-
lations. We now list the different points that need to be addressed to insure the accuracy of the retrieval
procedure.
1. First, our optical calculations (Eq. (5)) use Mie theory for spherical particles. This can introduce
some degrees of uncertainties in simulated LW spectra, especially near the resonant peaks (Legrand
et al., 2014). Since almost all climate models use Mie theory to calculate dust optical properties, we
decided to assume spherical particles at present. This assumption could be, however, not fully ap-
propriate for remote sensing applications, both ground-based and satellite, given that particle non-
sphericity is currently taken into account in many inversions algorithms (AERONET, POLDER,
etc.; e.g., Dubovik et al., 2006).
2. Second, as discussed in Sect. 2.2, measured dust spectra at wavelengths > 6 μm represent only dust
absorption, with minimal contribution from scattering. Dufresne et al. (2002) show that the contri-
bution of LW scattering from dust is quite important in the atmosphere, especially under cloudy
conditions. Therefore, the impact of neglecting the scattering contribution has to be assessed. The





retrieval procedure used in this study is nearly independent of whether dust extinction or only ab-
sorption is used. Indeed, the combination of Eq. (5) with the Lorentz formulation in Eq. (7a) and
(7b) ensures the retrieval of n-k couples that are theoretically correct (fulfilling the K-K relation-
ship), and the specific quantity to reproduce by Eq. (5) – i.e., extinction or absorption – provides
only a mathematical constraint on the retrieval. Therefore, neglecting the scattering contribution to
the LW spectra has no influence on the estimates of the refractive index, and the real and the imagi-
nary parts obtained in this study represent both the scattering and the absorption components of the
dust extinction.
3. Third, optical calculations are performed only at wavelengths > 6 µm, while in the range 2-6 µm
$k(\lambda)$ is fixed to the value obtained at 6 µm. We examine the accuracy of this assumption. Given that,
over the whole 2-6 µm range, dust is expected to have a negligible absorption (k is close to zero, see
Di Biagio et al., 2014a), fixing k at the value at 6 µm is a reasonable approximation. Concerning the
impact of this assumption on the retrieval of n, it should be pointed out that in the range 2-6 µm,
when k is very low, the shape of the n spectrum is determined only by the anchor point $n_{vis}$, and the
exact value of k is not relevant.

## 480    3.2 Uncertainty estimation

The uncertainty in the retrieved refractive index was estimated with a sensitivity analysis. Towards this
goal, n and k were also obtained by using as input to the retrieval algorithm the measured $\beta_{abs}(\lambda)$ and
size distribution ± their estimated uncertainties. The deviations of the values of n and k retrieved in the
sensitivity study with respect to those obtained in the first inversion were estimated. Then, we comput-
ed a quadratic combination of these different factors to deduce the uncertainty in n and k.
The results of the sensitivity study indicated that the measurement uncertainties on $\beta_{abs}(\lambda)$ (±10%) and
the size distribution (absolute uncertainty on the number concentration, ±20-70%) have an impact of
~10-20% on the retrieval of n and k.
Additionally, a sensitivity analysis was performed to test the dependence of the retrieved LW refrac-
tive index on the accuracy of the shape of the size distribution above 8 µm. As discussed in Sect. 2.3.2,
the size distribution $\left[dN/d\log D_g\right]_{CESAM}$ used for the optical calculations was measured between 0.1 and
8 µm based on SMPS, SkyGrimm, and WELAS data. However, it was extrapolated to larger sizes by



applying a lognormal mode fit for particle diameters >8 μm, where measurements were not available.
The extrapolation was set to reproduce the shape of the WELAS size distribution between $D_g$~3-4 and
8 μm. In the sensitivity study, n and k were also obtained by using two different size distributions as
input to the retrieval algorithm, in which the extrapolation curve at $D_g$>8μm was calculated by consid-
ering the WELAS data ± their estimated y-uncertainties. The results of the sensitivity study indicate
that a change of the extrapolation curve between its minimum and maximum may induce a variation of
less than 10% on the retrieved n and k.
The total uncertainty on n and k, estimated as the quadratic combination of these factors, was close to

20%.

An additional source of uncertainty linked to the size distribution, which however we do not quantify
here, concerns the choice of performing a single-mode extrapolation above 8 μm, which means ne-
glecting the possible presence of larger dust modes.

## 4. Selection of soil samples: representation of the dust mineralogical variability at the global scale

Nineteen soil samples were selected for experiments from a collection of 137 soils from various source
areas worldwide. Their location is shown in Fig. 2. The main information on the provenance of the
selected soils is summarized in Table 2. Soils were grouped in the nine regions identified by Ginoux et
al. (2012): Northern Africa, Sahel, Eastern Africa and Middle East, Central Asia, Eastern Asia, North
America, South America, Southern Africa, and Australia. The choice of the soils to analyze was per-
formed according to two criteria: 1) soils had to represent all major arid and semi-arid regions, as de-
picted by Ginoux et al. (2012) and 2) their mineralogy should envelope the largest possible variability
of the soil mineralogical composition at the global scale.
A large set of soils were available for Northern Africa, the Sahel, Eastern Africa and the Middle East,
Eastern Asia, and Southern Africa. Here, the selection was performed using as guidance the global
database of Journet et al. (2014), reporting the composition of the clay (<2 μm diameter) and silt (<60
μm diameter) fractions in terms of 12 different minerals. Amongst them, we analyzed the variability of
the minerals that are most abundant in dust as well as most optically relevant to LW absorption, name-
ly, illite, kaolinite, calcite, and quartz in the clay fraction, and calcite and quartz in the silt fraction. The



comparison of the extracted (from the Journet database) clay and silt compositions of the soils corresponding to the available samples resulted in the selection of five samples for Northern Sahara, three for the Sahel, three for Eastern Africa and the Middle East, and two for Eastern Asia and Southern Africa, as listed in Table 2.

For Northern Africa, we selected soils from the Northern Sahara (Tunisia, Morocco), richer in calcite and illite, Central Sahara (Libya and Algeria), enriched in kaolinite compared to illite and poor in calcite, and Western Sahara (Mauritania), richer in kaolinite. The three samples from the Sahel are from Niger, Mali and Chad (sediment from the Bodélé depression), and are enriched in quartz compared to Saharan samples. The selected soils from Northern Africa and the Sahel represent important sources for medium and long-range dust transport towards the Mediterranean (Israelevich et al., 2002) and the Atlantic Ocean (Prospero et al., 2002; Reid et al., 2003). In particular, the Bodélé depression is one of the most active sources at the global scale (Goudie and Middleton, 2001; Washington et al., 2003).

The three soils from Eastern Africa and the Middle East are from Ethiopia, Saudi Arabia, and Kuwait, which are important sources of dust in the Red and the Arabian seas (Prospero et al., 2002) and the North Indian Ocean (Leon and Legrand, 2003). These three samples differ in their content of calcite, quartz, and illite-to-kaolinite mass ratio (I/K).

For the second largest global source of dust, Eastern Asia, we considered two samples representative of the Gobi and the Taklimakan deserts, respectively. These soils differ in their content of calcite and quartz. Unfortunately, no soils are available for Central Asia, mostly due to the difficulty of sampling these remote desert areas.

For Southern Africa, we selected two soils from the Namib desert, one soil from the area between the Kuiseb and Ugab valleys (Namib-1) and one soil from the Damaraland rocky area (Namib-2), both sources of dust transported towards the South-Eastern Atlantic (Vickery et al. 2013). These two soils present different compositions in term of calcite content and I/K ratio.

A very limited number of samples were available in the soil collection for North and South America and Australia. These soils were collected in the Sonoran Desert for North America, in the Atacama and Patagonian deserts for South America, and in the Strzelecki desert for Australia. The Sonoran Desert is a permanent source of dust in North America, the Atacama desert is the most important source of dust in South America, whilst Patagonia emissions are relevant for long-range transport towards Antarctica





(Ginoux et al., 2012). The Strzelecki desert is the seventh largest desert of Australia. No mineralogical
criteria were applied to these areas.
A summary of the mineralogical composition of the nineteen selected soils is shown in Fig. 3 in com-
parison with the full range of variability obtained considering the full data from the different nine dust
source areas. As illustrated by this figure, the samples chosen for this study cover the entire global
variability of the soil compositions derived by Journet et al. (2014).

**5. Results**
**5.1 Atmospheric representativity: mineralogical composition**
The mineralogical composition of the nineteen generated aerosol samples as measured by XRD analy-
sis is shown in Fig. 4. The aerosol composition is dominated by clays (~55-95% for the different sam-
ples), with variable contents of quartz, calcite, dolomite, and feldspars. Identified clay species are: il-
lite, kaolinite, smectite, palygorskite, and chlorite. Illite and kaolinite are ubiquitous; smectite and
palygorskite are detected in some of the samples (Algeria, Ethiopia, Saudi Arabia, Kuwait, Arizona,
and both samples from Namibia); in contrast, chlorite is found only in the two Chinese and in the Chil-
ean samples. The estimated contribution of illite, kaolinite and chlorite to the total clay mass are shown
in Fig. 4 for Northern Africa (Algerian sample excluded, given that also smectite is detected in this
sample) and Eastern Asian aerosols. Quartz ranges from 2 to 32% by mass in the samples, with the
highest values measured for Australia, Patagonia, and Niger dust. Calcite is less than 17%, with maxi-
ma observed for Tunisia and Gobi dusts. Conversely, minor traces of dolomite (<2%) are detected in
all the different samples. Finally, feldspars (orthoclase and albite) represent less than 9% of the dust
composition.
Observations from the present study capture well the global tendencies of the dust mineralogical com-
positions as observed in several studies based on aerosol field observations, both from ground-based
and airborne samples (e.g., Sokolik and Toon, 1999; Caquineau et al., 2002; Shen et al., 2005 ; Jeong,
2008; Kandler et al., 2009; Scheuvens et al., 2013; Formenti et al., 2014). For instance, at the scale of
Northern Africa, we correctly reproduce the geographical distribution of calcite, which is expected to
be larger in Northern Saharan samples (Tunisia, Morocco), and very low or absent when moving to-
wards the Southern part of the Sahara and the Sahel (Libya, Algeria, Mauritania, Niger, Mali, and Bo-



délé samples). Similarly, we observe an increase of the aerosol quartz content from Northern Sahara
towards the Sahel, which is well known at the regional scale of Northern Africa (e.g., Caquineau et al.,
2002). Also, we identify the presence of chlorite in the Eastern Asian samples (Gobi and Taklimakan),
in agreement with field observations in this region (Shen et al., 2005). A more direct comparison of
our data with field measurements of the dust mineralogical composition is rather complicated due to
possible differences linked to the size distribution and representativeness of the specific sources be-
tween our data and field measurements (Perlwitz et al., 2015a, 2015b). For the Niger sample only,
however, a semi-quantitative comparison can be performed against field data of the dust mineralogy
obtained for aerosols collected at Banizoumbou during the AMMA (African Monsoon Multidiscipli-
nary Analysis) campaign in 2006. The mineralogy for these samples was provided by Formenti et al.
(2014). For a case of intense local erosion at Banizoumbou, they showed that the aerosol is composed
of 51% (by volume) of clays, 41% of quartz, and 3% of feldspars. Our Niger sample generated from
the soil collected at Banizoumbou, is composed of 64% of clays, 30% of quartz, and 5% of feldspars,
in relatively good agreement with the field observations.

**5.2 Atmospheric representativity: size distribution**
The size distribution of the dust aerosols measured at the peak of the dust injection in the chamber is
shown in Fig. 5. We report in the plot the normalized surface size distribution, defined as:
$$\frac{dS}{d \log D_g} (normalized) = \frac{1}{S_{tot}} \cdot \left( \frac{\pi}{4} D_g^2 \left[ dN / d \log D_g \right]_{CESAM} \right) \qquad (9)$$

with $S_{tot}$ the total surface area. The surface size distribution is the quantity that determines dust optical
properties (see Eq. 5). The dust surface size distributions present multimodal structures, where the
relative proportions of the different modes vary significantly between the samples. The dust mass con-
centration at the peak of the injection estimated from size distribution data varies between 2 and 310
mg m$^{-3}$. These values are comparable to what has been observed close to sources in proximity to dust
storms (Goudie and Middleton, 2006; Rajot et al., 2008; Kandler et al., 2009; Marticorena et al., 2010).
Given that the protocol used for soil preparation and aerosol generation is always the same for the dif-
ferent experiments, the observed differences in both the shape of the size distribution and the mass
concentration of the generated dust aerosols are attributed to the specific characteristics of the soils,
which may be more or less prone to produce coarse-size particles.





The comparison of the chamber data with observations of the dust size distribution from several air-
borne campaigns in Africa and Asia is shown in Fig. 6. This comparison suggests that the shape of the
size distribution in the chamber at the peak of the injection accurately mimic the dust distribution in
the atmosphere near sources.
The time evolution of the normalized surface size distribution within CESAM is shown in Fig. 7 for
two examples taken from the Algeria and Atacama experiments, while an example of the dust number
and mass concentration evolution over an entire experiment is illustrated in Fig. S5 (supplementary
material). As shown in Fig. 7, the dust size distribution strongly changes with time due to gravitational
settling: the coarse mode above 5 µm rapidly decreases, due to the larger fall speed at these sizes (~1
cm s$^{-1}$ at 10 µm, compared to ~0.01 cm s$^{-1}$ at 1 µm; Seinfeld and Pandis, 2006), and the relative im-
portance of the fraction smaller than $D_g$=5 µm increases concurrently. In the chamber we are thus able
to reproduce very rapidly (about 2 hours) the size-selective gravitational settling, a process that in the
atmosphere may takes about one to five days to occur (Maring et al., 2003). In order to compare the
dust gravitational settling in the chamber with that observed in the atmosphere the following analysis
was performed. For both Algeria and Atacama soils, the fraction of particles remaining in suspension
in the chamber as a function of time versus particle size was calculated as $dN_i(D_g)/dN_0(D_g)$, where
$dN_i(D_g)$ is the number of particles measured by size class at the i-time (i corresponding to 30, 60, 90
and 120 min after injection) and $dN_0(D_g)$ represents the size-dependent particle number at the peak of
the injection. The results of these calculations are shown in the lower panels of Fig. 7, where they are
compared to the fraction remaining airborne after 1-2 days obtained in the field study by Ryder et al.
(2013b) for mineral dust transported out of Northern Africa in the Saharan Air Layer (Karyampudi et
al., 1999), that is, at altitudes between 1.5 and 6 km above sea level. The comparison indicates that the
remaining particle fraction observed 30 minutes after the peak of the injection is comparable to that
obtained by Ryder et al. (2013b) for particles smaller than ~3 to 8 µm (depending on the soil) but that
the depletion is much faster for larger particles. This suggests, on the one hand, that the number frac-
tion of coarse particles in the chamber depends on the initial size distribution, that is, on the nature of
the soil itself. On the other hand, it shows the limitation of the four-blade fan in providing a vertical
updraft sufficient to counterbalance the gravimetric deposition for particles larger than about 8 µm.
This point, however, is not surprising since it is clear that in the laboratory it is not possible to repro-
duce the wide range of dynamical processes that occur in the real atmosphere, and so to obtain a faith-
ful reproduction of dust gravitational settling and the counteracting re-suspension mechanisms. None-





theless, it should be noted that the rate of removal is higher at the earlier stage of the experiments than
towards their end. The size-dependent particle lifetime, defined as the value at which $dN/dN_0$ is equal
to $1/e$ (McMurry and Rader, 1985), is relatively invariant for particles smaller than $D_g < \sim 2$ μm (> 60
min). This indicates that no significant distortion of the particle size distribution occurs after the most
significant removal at the beginning of the experiment, and that the fine-to-coarse proportions are
modified with time in a manner consistent with previous field observations on medium- to long-
transport (e.g., Maring et al., 2003; Rajot et al., 2008; Reid et al., 2008; Ryder et al., 2013b; Denjean et
al., 2016).

**5.3 Dust LW extinction and complex refractive index spectra for the different source regions**
Figure 8 shows the dust LW spectral extinction coefficients measured at the peak of the injection for
the nineteen aerosol samples. As discussed in Sect. 2.2, the spectra in Fig. 8 show the contribution of
dust scattering below 6 μm, while the absorption spectrum only is measured above 6 μm. In this wave-
length range, significant differences are observed when comparing the samples, which in turn are
linked to differences in their mineralogical composition.
Figure 8 allows the identification of the spectral features of the minerals presenting the strongest ab-
sorption bands, in particular in the 8-12 μm atmospheric window (Table 3). The most prominent ab-
sorption peak is found around 9.6 μm for all samples, where clays have their Si—O stretch resonance
peak. The shape around the peak differs according to the relative proportions of illite and kaolinite in
the samples, as is illustrated with the results for Tunisia, Morocco, Ethiopia, Kuwait, Arizona, Patago-
nia, Gobi and Taklimakan samples (richer in illite) compared to Libya, Algeria, Mauritania, Niger,
Bodélé, Saudi Arabia, and Australia (richer in kaolinite). Aerosols rich in kaolinite also show a sec-
ondary peak at ~10.9 μm. The spectral signature of quartz at 9.2 and 12.5-12.9 μm is ubiquitous, with
a stronger contribution in the Bodélé, Niger, Patagonia, and Australia samples. Aerosols rich in calcite,
such as the Tunisia, Morocco, Saudi Arabia, Taklimakan, Arizona, Atacama, and Namib-1 samples
show absorption bands at ~7 and 11.4 μm. Conversely, these are not present in the other samples and
in particular in none of the samples from the Sahel. Finally, the contribution of feldspars (albite) at 8.7
μm is clearly detected only for the Namib-1 sample.
The intensity of the absorption bands depend strongly on the particle size distribution, in particular on
the contribution of the aerosol super-micron fraction, as well as on the total dust mass concentration.





These, as discussed in the previous section, are associated with the specific characteristics of each of
the soils used and their propensity for dust emission. The highest values of dust absorption that can be
seen in Fig. 8 for the 8–12 µm spectral region appear for the Bodélé aerosol sample. In this particular
sample, the super-micron particles represent 45% of the total particle number at the peak of the injec-
tion, and this sample showed the highest mass concentration in the chamber (310 mg m$^{-3}$). Conversely,
the lowest absorption is measured for the aerosols from Mauritania, Mali, Kuwait, and Gobi, for which
the super-micron particle fraction and the mass concentrations are lower.
The intensity of the spectral extinction rapidly decreases after injection, following the decrease of the
super-micron particle number and mass concentration. As an example, Fig. 9 shows the temporal evo-
lution of the measured extinction spectrum for the Algeria and Atacama aerosols. The intensity of the
absorption band at 9.6 µm is about halved after 30 min and reduced to ~20-30% and <10% of its initial
value after 60 min and 90-120 min, respectively. Because of the size-dependence of the mineralogical
composition, notably the relative proportions of quartz and calcite with respect to clays (Pye et al.,
1987), settling could also modify the spectral shape of the extinction spectrum. This effect was inves-
tigated for the two example cases, Algeria and Atacama, by looking at the temporal evolution of the
ratios of the measured extinction coefficient in some specific mineral absorption bands. Changes
would indicate that the time variability of the mineralogical composition is optically significant. For
the Algeria case, we have considered the quartz (12.5 µm) versus clay (9.6 µm) bands, and for the
Atacama case the calcite (~7 µm) versus clay (9.6 µm) bands. For both cases, the calculated ratios do
not change significantly with time, i.e. they agree within error bars: for Algeria, the quartz-to-clay ratio
is 0.21±0.03 at the peak of the injection and 0.25±0.04 120 min later; for Atacama, the calcite-to-clay
ratio is 0.73±0.10 and 0.67±0.09 for the same times. Similar results were also obtained for the other
samples, with the only exception of Saudi Arabia and Morocco for which we observed an increase of
the calcite-to-clay ratio with time. The time invariance of the quartz-to-clays and calcite-to-clays ratios
observed for the majority of the analyzed aerosol samples agrees with the observations of the size-
dependent dust mineralogical composition obtained by Kandler et al. (2009). These authors showed
that in the super-micron diameter range up to ~25 µm, i.e. in the range where dust is mostly LW-
active, the quartz/clay and calcite/clay ratios are approximately constant with size. This would suggest
that the loss of particles in this size range should not modify the relative proportions of these minerals,
and thus their contributions to LW absorption. Nonetheless, the different behavior observed for Saudi



Arabia and Morocco would possibly indicate differences in the size-dependence of the mineralogical
composition compared to the other samples.
For each soil, the estimated real (n) and imaginary (k) parts of the complex refractive index are shown
in Fig. 10. The reported n and k correspond to the mean of the 10-min values estimated between the
peak of the injection and 120 min later. This can be done because, for each soil, the time variation of
the complex refractive index is moderate. Standard deviations, not shown in Fig. 10 for the sake of
visual clarity, are <10% for n and <20% for k. Figure 10 shows that the dust refractive index widely
varies both in magnitude and spectral shape from sample to sample, following the variability of the
measured extinction spectra. The real part n varies between 0.84 and 1.94, while the imaginary part k
is between ~0.001 and 0.92.

**6. Discussion**
**6.1 Predicting the dust complex refractive index based on its mineralogical composition**
Our results show that the LW refractive index of mineral dust having different mineralogical composi-
tions varies considerably. Nevertheless, at wavelengths where the absorption peaks due to different
minerals do not overlap, this variability can be predicted from the composition-resolved mass concen-
trations. These considerations are illustrated in Fig. 11, where we relate the mean values of the dust k
in the calcite, quartz, and clay absorption bands between 7.0 and 11.4 μm to the percent mass fraction
of these minerals in the dust. Mean k values were calculated as averages over the filter sampling times.
For calcite and quartz (resonance peaks at 7.0, 9.2, and 11.4 μm), this relation is almost linear. These
two minerals are commonly large in grain size and well crystallized. Their quantification by XRD is
certain and they produce a strong and well-identified absorption peak in the LW. Nonetheless, there
seems to be a lower limit of the percent mass of calcite (around 5%) that gives rise to absorption at 7
μm, and therefore measurable k-values (Fig. 11). Conversely, at 11.4 μm, non-zero k-values are ob-
tained even in the absence of calcite, due to the interference of the calcite peak and the clay resonance
bands.
Poorer or no correlation is found between k and the percent mass fraction in the absorption bands of
clays at 9.6 and 10.9 μm. This different behavior is not unexpected. Clay minerals such as kaolinite,
illite, smectite and chlorite are soil weathering products containing aluminium and silicon in a 1:1 or



1:2 ratio (tetrahedral or octahedral structure, respectively). As a consequence, the position of their vi-
brational peaks is very similar (Dorschner et al., 1978; Querry, 1987, Glotch et al., 2007). In the at-
mosphere, these minerals undergo aging by gas and water vapor adsorption (Usher et al., 2003; Schut-
tlefield et al., 2007). As a result of the production conditions in the soils (weathering) and aging in the
atmosphere, their physical and chemical conditions (composition, crystallinity, aggregation state)
might differ from one soil to the other, and from that of mineralogical standards. That is the reason
why XRD measurements of clays in natural dust samples might be erroneous, and why we prefer to
estimate the clay fraction indirectly. Nonetheless, the indirect estimate is also prone to error, and de-
pends strongly on an independent estimate of the total mass (which, in the presence of large particles
can be problematic) as well as the correct quantification of the non-clay fraction. This is likely reflect-
ed in the large scatter observed in Fig. 11 when trying to relate the k-value distribution to the corre-
sponding percent mass of clays. These considerations also affect the speciation of clays, and explain
the similar results obtained when separately plotting the spectral k-values against the estimate kaolinite
or illite masses. The superposition of the resonance bands of these two clays, as well as those of the
smectites, which in addition are often poorly crystallized and therefore difficult to detect by XRD, as
well as those in the quartz absorption band at 9.2 μm, suggests that a more formal spectral deconvolu-
tion procedure based on single mineral reference spectra is needed to understand the shape and magni-
tude of the refractive index in this spectral band.

## 6.2 Dust complex refractive index versus size distribution during atmospheric transport

Quantifying the radiative impact of dust depends not only on the ability to provide spatially-resolved
optical properties, but also on the accurate representation of the possible changes of these properties
during transport. In the LW, this effect is amplified by the changes in the size distribution, particularly
the loss of coarse particles. Our experiments accurately capture the overall features of the dust size
distribution, including the extent and modal position of the coarse particle mode. However, the deple-
tion rate with time for coarse particles is higher than observed in the atmosphere (e.g., Ryder et al.,
2013b). The size distribution after 30 minutes still contains a significant, relatively invariant, but not-
predictable fraction of coarse particles. This calls for two considerations: 1) the refractive indices ob-
tained at the early stage of the experiments (within 30 minutes after the dust injection) are representa-
tive of dust at short to medium ranges of transport; 2) the refractive indices after 30 minutes of dura-



tion are likely to represent long-range transported dust still containing coarse particles in a fraction that
will depend on the original soil. In our study, the calculated refractive indices do not change with time
in parallel with the observed changes in the size distribution, thus suggesting that a constant value can
be assumed close to the source and during transport. Still, further experiments taking into account only
the fine fraction of the aerosols will be needed to constrain the size-dependence of the refractive index.

**6.3 Comparison with the literature**
In Fig. 12, we compare our results with estimates of the dust refractive index reported in the literature.
We consider data by Volz (1972, 1973) for dust collected in Germany and at Barbados, Fouquart et al.
(1987) for Niger sand, and Di Biagio et al. (2014a) for dust from Algeria and Niger. We also report
data for dust as assumed in the OPAC database (Optical Properties of Aerosols and Clouds; Hess et al.,
1998; Koepke et al., 2015), one of the most frequently used references in climate modeling and remote
sensing applications. Because of the limited regional span, the literature data clearly cannot do justice
to the full range of magnitude and of the spectral variability of the LW complex refractive index that is
presented in our dataset. In particular, clearly none of the published data represent the contribution of
calcite at ~7 μm. Some of the data (Volz, 1973; Fouquart et al., 1987; OPAC) overestimate k above 11
μm, where the 12.5-12.9 μm quartz absorption band is found. The best correspondence, especially
above 10 μm, is found with Di Biagio et al. (2014a). In the 8–12 μm atmospheric window, the agree-
ment with our estimated mean value is moderate, but the range of variability around the mean and its
spectral dependence are underrepresented. A shift towards larger wavelengths is also observed for the
main clay absorption peak at ~9.6 μm for Volz (1973) and Di Biagio et al. (2014a), which is possibly
linked to the different method used in these studies to retrieve the complex refractive index (pellet
spectroscopy approach) compared to our data. The agreement is even less satisfactory for the real part
of the refractive index (upper panel of Fig. 12), which is overestimated in OPAC and Volz (1973) and
underestimated in Fouquart et al. (1987). As discussed in Di Biagio et al. (2014a), differences for the
real part between the various studies come mostly from the different methods used to estimate the dust
refractive index. The methods used in the literature most often do not fulfil the Kramers-Kronig rela-
tionship for the n-k couples. The only dataset that fulfils the Kramers-Kronig relationship is Fouquart
et al. (1987), but that has the drawback of underestimating n as a consequence of the low value of $n_{vis}$
(~1) assumed in the retrieval.





## 7. Conclusions and perspectives

In this study we have presented a new set of laboratory in situ measurements of the LW extinction spectra and complex refractive indices of mineral dust aerosols from nineteen natural soils from source regions in Northern Africa, Sahel, Middle East, Eastern Asia, North and South America, Southern Africa, and Australia. These sources are representative of the heterogeneity of the dust composition at the global scale. Consequently, the envelope of refractive index data obtained in this study can adequately represent the full range of variability for dust as function of the global variability of its mineralogical composition. These data are expected to be widely applicable for both radiative transfer modelling and remote sensing applications.

The experiments described here were conducted in the realistic and dynamic environment of the 4.2 $m^3$ CESAM chamber. Dust aerosols generated in the chamber are characterized by a realistic size distribution, including both the sub-micron and the super-micron fraction, and they have an atmospherically representative mineralogical composition, including the main LW active minerals, such as quartz, clays, and calcite. The complex refractive index of dust at LW wavelengths is obtained following a rigorous approach that permits to determine n-k couples that satisfy the Kramers-Kronig relation. Refractive index data from the present study are much more reliable than those provided by DB14, given that a better estimate of $n_{vis}$ was used in the retrieval algorithm. The average uncertainty in the obtained n and k is ~20%.

The main results from this work can be summarized as follows.

1. The LW refractive index of dust varies strongly both in magnitude and spectral shape as a result of the variability of the particle mineralogy related to the source region of emission. The available literature data (Volz, 1972, 1973; Fouquart et al., 1987; OPAC, Hess et al., 1998, Koepke et al., 2015) used nowadays in climate models and satellite retrievals, do not adequately represent either the magnitude, or the spectral features and the variability of the LW refractive index of mineral dust observed in our dataset. In consequence, we recommend the use of source-specific extinction spectra/refractive indices rather than generic values.

2. We observe a linear relationship between the magnitude of the LW refractive index and the mass concentration of specific minerals, i.e., quartz and calcite. This opens the possibility of providing predictive relationships to estimate the LW refractive index of dust at specific bands based on an assumed or predicted mineralogical composition, or conversely, to estimate the dust composition

(even partially) from measurements of LW extinction at specific wavebands. This could have im-
portant implications for the representation of LW optical properties of dust in climate models,
which have started to incorporate the representation of dust mineralogy in their schemes (Scanza et
al., 2015; Perlwitz et al., 2015a). In addition, the possibility to relate the mass of minerals to the ab-
sorption at specific bands implies that the LW extinction spectra measured from space can be used
to distinguish between different dust sources.
3. The spectral shape of the dust extinction spectrum does not seem to change significantly with time
due to the loss of coarse particles by gravitational settling. This suggests that, despite the dust
coarse mode being increasingly depleted, the relative proportions of minerals do not change signifi-
cantly with time or at least that their changes do not affect the overall optical response of the dust
samples. In consequence, the retrieved LW refractive index does not change, and therefore can be
used to represent short-to-medium range transport conditions. This finding supports the common
practice in global models to treat the dust LW refractive index as static during transport. This also
implies that to represent the dust LW refractive index vs mineralogy, models just have to reproduce
the dust composition at the source, without the necessity of following its changes during transport,
which could be a challenge. This would considerably simplify the representation of dust mineralogy
in models.
The unique dataset presented in this study should be particularly useful for improving the dust-climate
interactions within regional and global models, and to take into account the geographical variability of
the dust LW refractive index, which at present is not represented. This will allow obtaining a more
realistic representation of the dust LW effect and its radiative forcing upon climate. To date, as evi-
denced in Boucher et al. (2013), the sign of the dust direct effect remains unknown. In this regard, in
particular, we estimate lower dust absorption than in OPAC (see k curves in Fig. 12). The integral of
the OPAC dust refractive index (imaginary part) between 3 and 15 μm is about 20% larger compared
to the integral obtained from our max k curve; up to about one order of magnitude overestimate is
found when the integral of the OPAC k over the 3-15 μm range is compared to the integral of our min
k curve. In consequence of this, we can conclude that the use of OPAC data may introduce a systemat-
ic bias in modelling dust radiative effects at LW wavelengths.





The use of data from the present study also will help reducing uncertainties in satellite retrievals, thus
contributing to improve the remote sensing capability over regions affected by dust (e.g., Capelle et al.,
2014; Cuesta et al., 2015).
The work presented in this paper also opens various perspectives.
First, the results of the present study clearly suggest that the LW refractive index of dust varies at the
regional scale, as can be observed in Fig. 10 for Northern Africa, Sahel, the Middle East, Eastern Asia,
South America and Southern Africa. This regional variability has to be characterized further in order to
better assess the influence of dust on regional climate.
Second, the possibility of a more formal spectral deconvolution procedure based on single mineral
reference spectra to understand the shape, magnitude, and temporal variability of the refractive index
in all different spectral bands must be investigated. This could strongly help finding robust relation-
ships linking the dust refractive index to the particle mineralogy.
Third, further experimental efforts by increasing the lifetime and selecting size classes will be needed
to verify better the applicability of the obtained refractive index to long-range transport conditions.
Also, the experiments described here were done in conditions when dry deposition is the only aging
process. Other aging processes, such as heterogeneous reactions, mixing with other aerosol types, or
water uptake, have to be investigated to evaluate their impact on the LW refractive index during
transport. For instance, some studies suggest a possible enhancement of dust LW absorption over spe-
cific bands if water uptake occurs (Schuttlefield et al., 2007) or if dust mixes with soot (Hansell et al.,

2011).


**Appendix 1. Control experiment with ammonium sulfate particles**
In order to validate the methodology applied in this study, a control experiment was performed on
ammonium sulfate aerosols. Particles were generated from a 0.03 M solution of ammonium sulfate
using a constant output atomizer (TSI, model 3075). The aerosol flow passed through a diffusion drier
(TSI, model 3062), to be then injected in the CESAM chamber at a flow of 10 L min$^{-1}$ for 10 minutes.
At the peak of the injection the aerosol concentration reached ~160 $\mu$g m$^{-3}$ and the size distribution
was mono-modal and centered at ~0.06 $\mu$m. The LW spectrum of ammonium sulfate measured in
CESAM at the peak of the injection is shown in Fig. A1 for the 2-15 $\mu$m range. Absorption bands at-
tributed to gas-phase water vapor and $CO_2$ present in the chamber during the experiments are indicated



in the plot. The 2-15 μm spectral region includes three of the four active vibrational modes of the am-
monium sulfate salt: $v_3(NH_4^+)$ (3230 cm$^{-1}$ or 3.10 μm), $v_4(NH_4^+)$ (1425 cm$^{-1}$ or 7.02 μm; not identified
in the plot due to its superposition with the water vapor band), and $v_3(SO_2^{-4})$ (1117 cm$^{-1}$ or 8.95 μm).
The $v_4(SO_2^{-4})$ is at 620 cm$^{-1}$ (16.12 μm), thus below the measurement range of the FTIR. The retrieval
algorithm described in Sect. 3 was applied to estimate the complex refractive index of ammonium sul-
fate aerosols. Calculations were performed only in the 8-10 μm range where the $v_3(SO_2^{-4})$ band is
found and where the contamination by water vapor is minimal. The value of $n_{vis}$ to use as input to the
algorithm was set at 1.55, based on the analysis of simultaneous SW optical data (not discussed here).
The results of the calculations are shown in Fig. A1. The comparison with the optical constants pro-
vided by Toon et al. (1976), also shown in Fig. A1, is very satisfactory. A small bias is observed for
our retrieved n compared to the values by Toon et al. (1976). This can be possibly linked to the method
used in Toon et al. (1976) to retrieve the real part of the refractive index, which is based on the meas-
urement of the normal incident reflectivity of a bulk sample instead of absorption data of aerosol parti-
cles, as in our experiments. Overall, the results of the control experiment indicate that the CESAM
approach and the proposed retrieval algorithm allow to reproduce the LW spectral signature of the
aerosols and to estimate accurately their complex refractive index.















**Author contributions**


C. Di Biagio, P. Formenti, Y. Balkanski, and J. F. Doussin designed the experiments and discussed the
results. C. Di Biagio realized the experiments and performed the full data analysis with contributions
by P. Formenti, L. Caponi, M. Cazaunau, E. Pangui, S. Caquineau, and J.F. Doussin. S. Nowak per-
formed the XRD measurements. M. O. Andreae, K. Kandler, T. Saeed, S. Piketh, D. Seibert, and E.
Williams collected the soil samples used for experiments. E. Journet participated to the selection of the
soil samples for experiments and contributed to the scientific discussion. C. Di Biagio, P. Formenti,
and Y. Balkanski wrote the manuscript with comments from all co-authors.

**Acknowledgements**


This work was supported by the French national programme LEFE/INSU, by the EC within the I3 pro-
ject "Integration of European Simulation Chambers for Investigating Atmospheric Processes" (EU-
ROCHAMP-2, contract no. 228335), by the OSU-EFLUVE (Observatoire des Sciences de l'Univers-
Enveloppes Fluides de la Ville à l'Exobiologie) through dedicated research funding, and by the CNRS-
INSU supporting CESAM as national facility. C. Di Biagio was supported by the CNRS via the Labex
L-IPSL. K. Kandler received support from the German Science Foundation DFG (KA 2280/2). Field
sampling in Saudi Arabia was supported by a grant from King Saud University. The authors strongly
thank S. Alfaro, B. Chatenet, M. Kardous, R. Losno, B. Marticorena, J. L. Rajot, and G. Vargas, who
participated in the collection of the soil samples from Tunisia, Niger, Atacama, Patagonia, and the Go-
bi desert used in this study. The authors wish also to acknowledge J.L Rajot for his helpful comments.



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





**Tables**
**Table 1.** Measured and retrieved quantities and their estimated uncertainties. For further details refer
to Sect. 2.

| | Parameter | Uncertainty | Uncertainty calculation |
|---|---|---|---|
| Optical LW | Transmission 2-16 μm, T | <10% | Quadratic combination of noise (~1%) and standard deviation over 10-min (5-10%) |
| | Extinction coefficient 2-16 μm, $\beta_{ext}(\lambda) = \dfrac{-\ln(T(\lambda))}{x}$ | ~10% | Error propagation formula[1] considering uncertainties on the measured transmission T and the optical path x (~2%) |
| Size distribution | SMPS geometrical diameter (D$_g$), $D_g = D_m / \chi$ | ~6% | Error propagation formula[1] considering the uncertainty on the estimated shape factor χ (~6%) |
| | SkyGrimm geometrical diameter (D$_g$) | <15.2% | Standard deviation of the D$_g$ values obtained for different refractive indices values used in the optical to geometrical conversion |
| | WELAS geometrical diameter (D$_g$) | ~5-7% | The same as for the SkyGrimm |
| | $\left[dN/d\log D_g\right]_{Corr,WELAS} = \left[dN/d\log D_g\right] / \left[1 - L_{WELAS}(D_g)\right]$ | ~20-70% | Error propagation formula[1] considering the dN/dlogD$_g$ st. dev. over 10-min and the uncertainty on L$_{WELAS}$ (~50% at 2 μm, ~10% at 8 μm) |
| | $\left[dN/d\log D_g\right]_{filter} = \left[dN/d\log D_g\right]_{CESAM} * \left[1 - L_{filter}(D_g)\right]$ | ~25-70% | Error propagation formula[1] considering the uncertainties on (dN/dlogD$_g$)$_{CESAM}$ and L$_{filter}$ (~55% at 2 μm, ~10% at 12 μm) |
| Mineralogical composition | Clays mass ( $m_{Clay} = M_{total} - m_Q - m_F - m_C - m_D - m_G$ ) | 8-26% | Error propagation formula[1] considering the uncertainty on M$_{total}$ (4-18%) and that on m$_Q$, m$_F$, m$_C$, m$_D$, and m$_G$ |
| | Quartz mass ( $m_Q = S_Q / K_Q$ ) | 9% | Error propagation formula[1] considering the uncertainty on the DRX surface area S$_Q$ (~2%) and K$_Q$ (9.4%) |
| | Feldspars mass ( $m_F = S_F / K_F$ ) | 14% (orthose), 8% (albite) | The same as for the quartz, K$_F$ uncertainty 13.6% (orthose) and 8.4% (albite) |
| | Calcite mass ( $m_C = S_C / K_C$ ) | 11% | The same as for the quartz, K$_C$ uncertainty 10.6% |
| | Dolomite mass ( $m_D = S_D / K_D$ ) | 10% | The same as for the quartz, K$_D$ uncertainty 9.4% |
| | Gypsum mass ( $m_G = S_G / K_G$ ) | 18% | The same as for the quartz, K$_G$ uncertainty 17.9% |

[1] $\sigma_f = \sqrt{\sum\limits_{i=1}^{n}\left(\dfrac{\partial f}{\partial x_i}\sigma_{x_i}\right)^2}$





**Table 2.** Summary of information on the soil samples used in this study.


| Sample name | Collection Coordinates | Geographical zone | Country | Desert zone |
|---|---|---|---|---|
| Tunisia | 33.02°N, 10.67°E | Northern Africa | Tunisia | Saharan desert (Maouna) |
| Morocco | 31.97°N, 3.28°W | Northern Africa | Morocco | Saharan desert (east of Ksar Sahli) |
| Libya | 27.01°N, 14.50°E | Northern Africa | Libya | Sahara desert (Sebha) |
| Algeria | 23.95°N, 5.47°E | Northern Africa | Algeria | Saharan desert (Ti-n-Tekraouit) |
| Mauritania | 20.16°N, 12.33°W | Northern Africa | Mauritania | Saharan desert (east of Aouinet Nchir) |
| Niger | 13.52°N, 2.63°E | Sahel | Niger | Sahel (Banizoumbou) |
| Mali | 17.62°N, 4.29°W | Sahel | Mali | Sahel (Dar el Beida) |
| Bodélé | 17.23°N, 19.03°E | Sahel | Chad | Bodélé depression |
| Ethiopia | 7.50°N, 38.65°E | Eastern Africa and the Middle East | Ethiopia | Lake Shala National Park |
| Saudi Arabia | 27.49°N, 41.98°E | Eastern Africa and the Middle East | SaudiArabia | Nefud desert |
| Kuwait | 29.42°N, 47.69°E | Eastern Africa and the Middle East | Kuwait | Kuwaiti desert |
| Gobi | 39.43°N, 105.67°E | Eastern Asia | China | Gobi desert |
| Taklimakan | 41.83°N, 85.88°E | Eastern Asia | China | Taklimakan desert |
| Arizona | 33.15 °N, 112.08°W | North America | Arizona | Sonoran desert |
| Atacama | 23.72°S, 70.40°W | South America | Chile | Atacama desert |
| Patagonia | 50.26°S, 71.50°W | South America | Argentina | Patagonian desert |
| Namib-1 | 21.24°S, 14.99°E | Southern Africa | Namibia | Namib desert (area between the Kuiseb and Ugab valleys) |
| Namib-2 | 19.0°S, 13.0°E | Southern Africa | Namibia | Namib desert (Damaraland, rocky area in north-western Namibia) |
| Australia | 31.33°S, 140.33°E | Australia | Australia | Strzelecki Desert |





**Table 3.** Position of LW absorption band peaks (6-16 µm) for the main minerals composing dust.
Montmorillonite is taken here as representative for the smectite family. For feldspars literature data are
available only for albite.

| Mineral species | Wavelength (µm) | Reference |
|---|---|---|
| Illite | 9.6 | Querry (1987) |
| Kaolinite | 9.0, 9.6, 9.9, 10.9 | Glotch et al. (2007) |
| Montmorillonite | 9.0, 9.6 | Glotch et al. (2007) |
| Chlorite | 10.2 | Dorschner et al. (1978) |
| Quartz | 9.2, 12.5-12.9 | Peterson and Weinman (1969) |
| Calcite | 7.0, 11.4 | Long et al. (1993) |
| Gypsum | 8.8 | Long et al. (1993) |
| Albite | 8.7, 9.1, 9.6 | Laskina et al. (2012) |




**Figures**



**Figure 1.** Schematic configuration of the CESAM set up for the dust experiments. The dust generation
(vibrating plate, Büchner flask containing the soil sample) and injection system is shown in the bottom
on the right side. The position of the SMPS, WELAS, and SkyGrimm used for measuring the size dis-
tribution, FTIR spectrometer, SW optical instruments, and filter sampling system are also indicated.

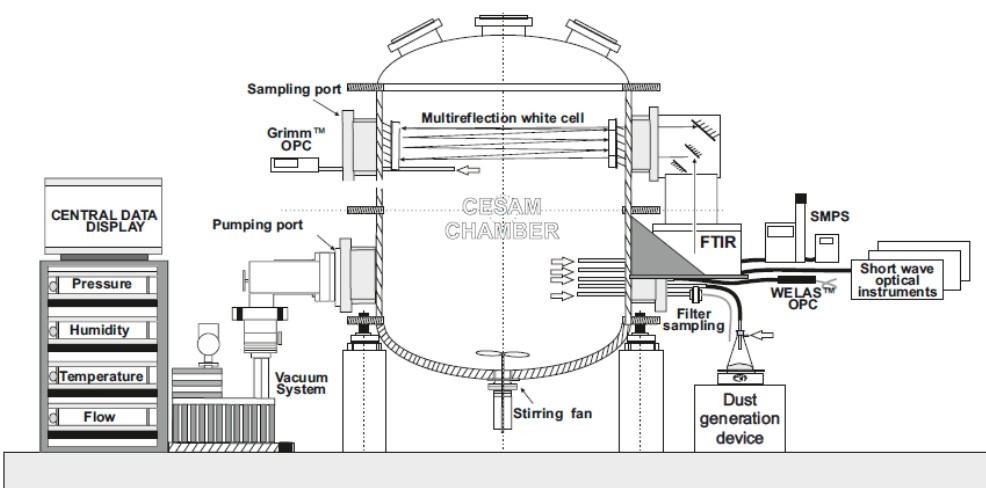
















**Figure 2.** Location (red stars) of the soil and sediment samples used to generate dust aerosols. The
nine yellow rectangles depict the different global dust source areas as defined in Ginoux et al. (2012):
1) Northern Africa, 2) Sahel, 3) Eastern Africa and Middle East, 4) Central Asia, 5) Eastern Asia, 6)
North America, 7) South America, 8) Southern Africa, and 9) Australia.

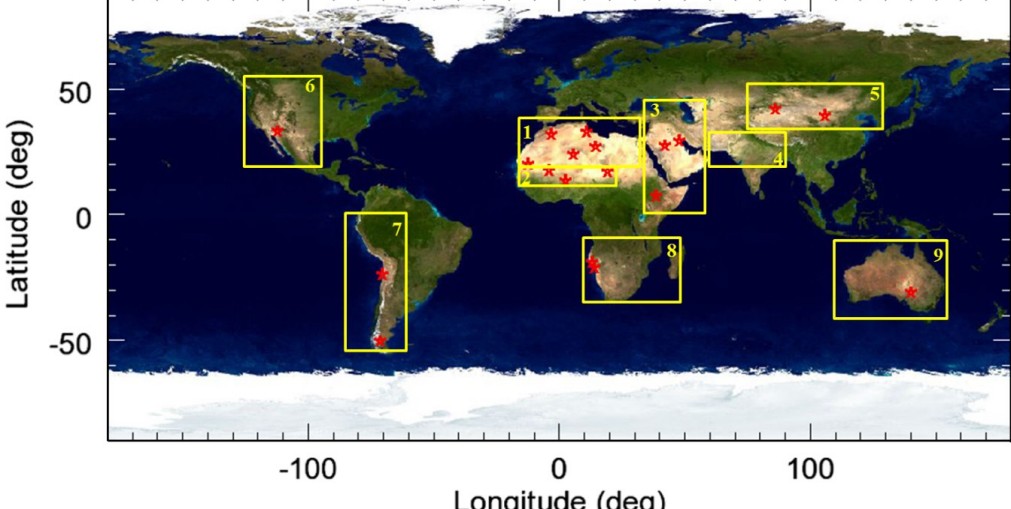



















**Figure 3.** Box and whisker plots showing the variability of the soil composition in the clay and silt fractions at the global scale, i.e., by considering all data from the nine dust source areas identified in Fig. 2. Data are from the soil mineralogical database by Journet et al. (2014). Dots indicate specific mineralogical characteristics (illite-to-kaolinite mass ratio, I/K, calcite and quartz contents, extracted from Journet et al.) of the soils used in the CESAM experiments, as listed in Table 2.

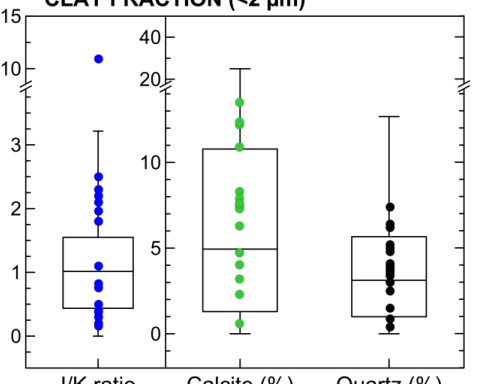

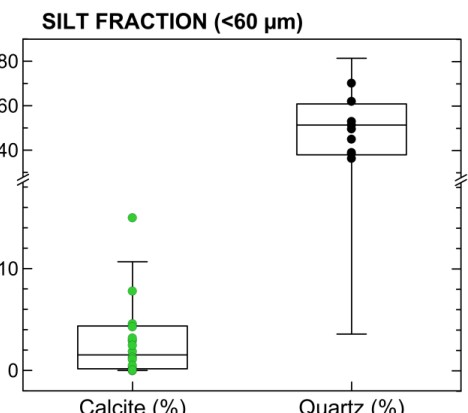



**Figure 4.** Mineralogy of the nineteen generated aerosol samples considered in this study, as obtained from XRD analysis. The mass apportionment between the different clay species (illite, kaolinite, chlorite) is shown for Northern African (Tunisia, Morocco, Libya, Mauritania, Niger, Mali, Bodélé) and Eastern Asian (Gobi, Taklimakan) aerosols based on literature compiled values of the illite-to-kaolinite (I/K) and chlorite-to-kaolinite (Ch/I) mass ratios (Scheuvens et al., 2013; Formenti et al. 2014). For all the other samples only the total clay mass is reported.

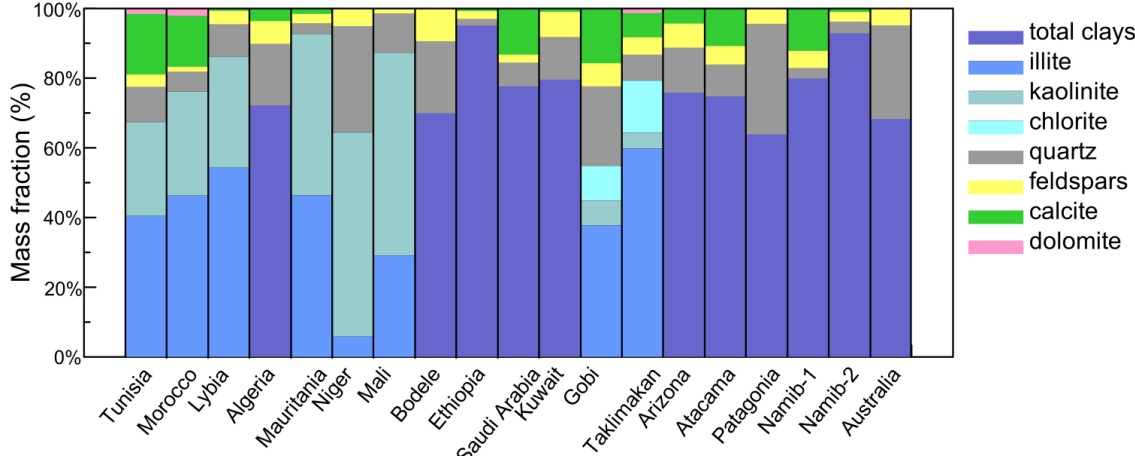



**Figure 5.** Surface size distributions in the CESAM chamber at the peak of dust injection for all cases analysed in this study; the total measured dust mass concentration and the percentage of the super-micron to sub-micron number fraction at the peak are also reported in the legend.



**Figure 6.** Comparison of CESAM measurements with dust size distributions from several airborne field campaigns in Africa and Asia. The grey shaded area represents the range of sizes measured in CESAM during experiments with the different samples. Data from field campaigns are: AMMA (Formenti et al., 2011), SAMUM-1 (Weinzierl et al., 2009), FENNEC (Ryder et al., 2013a), and ACE-Asia (Clarke et al., 2004). Min and max for the same data correspond to the range of variability observed for the campaigns considered.

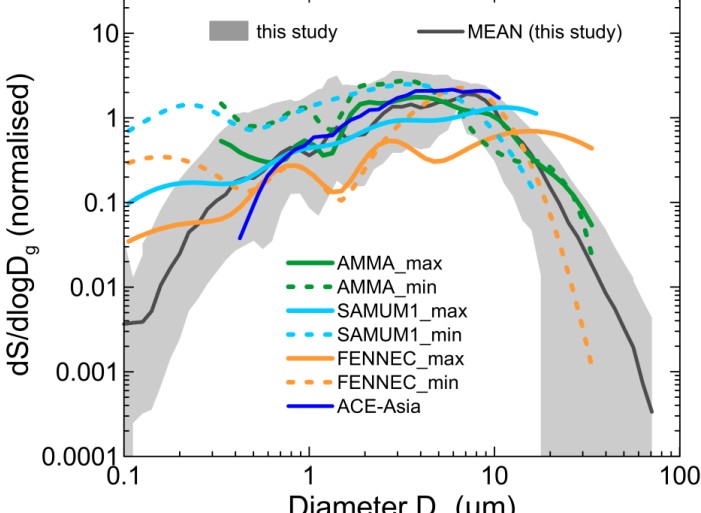





**Figure 7.** Upper panel: surface size distribution measured at the peak of the dust injection and at 30, 60, 90, and 120 minutes after injection for Algeria and Atacama aerosols. The dust mass concentration is also indicated in the plot. Lower panel: fraction of particles remaining airborne in the chamber as a function of time versus particle size calculated as $dN_i(D_g)/dN_0(D_g)$, where $dN_i(D_g)$ is the number of particles measured by size class at the i-time (i corresponding to 30, 60, 90 and 120 min after injection) and $dN_0(D_g)$ represents the size-dependent particle number at the peak of the injection. Values are compared to the estimate of Ryder et al. (2013b) for Saharan dust layers aged 1-2 days after emission.

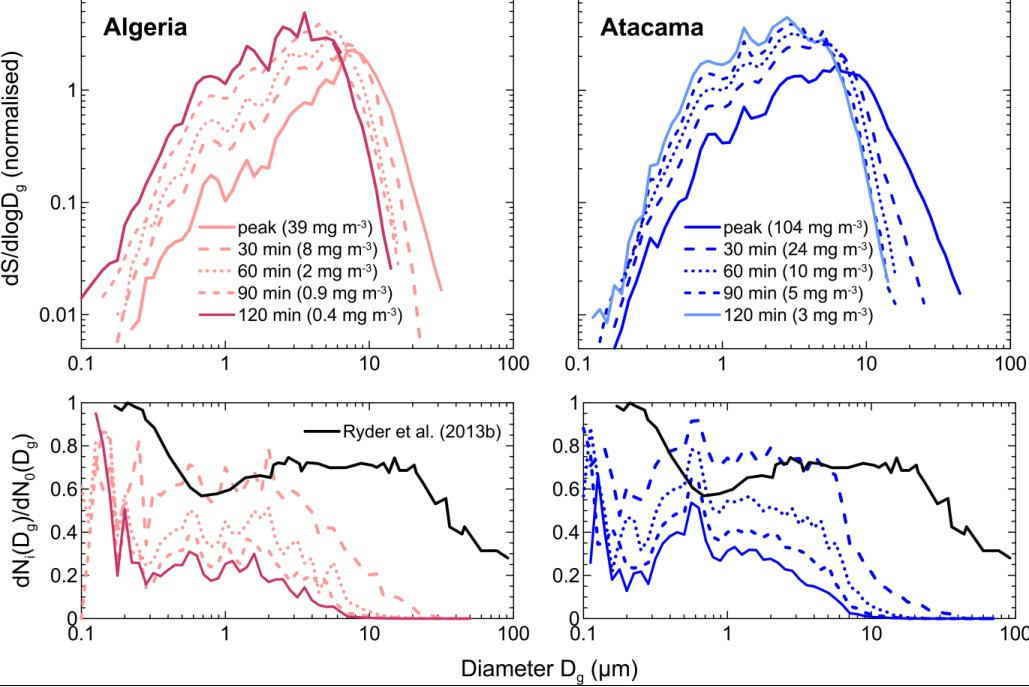



**Figure 8.** Dust extinction coefficient measured in the LW spectral range for the nineteen aerosol samples analysed in this study. Data for each soil refer to the peak of the dust injection in the chamber. Note that the y-scale is different for Northern Africa – Sahara compared to the other cases. Main absorption bands by clays at 9.6 µm, quartz (Q) at 9.2 and 12.5-12.9 µm, kaolinite (K) at 10.9 µm, calcite (C) at 7.0 and 11.4 µm, and feldspars (F) at 8.7 µm are also indicated in the spectra.



**Figure 9.** Extinction spectra measured at the peak of the dust injection and at 30, 60, 90, and 120
minutes after injection for Algeria and Atacama aerosols.


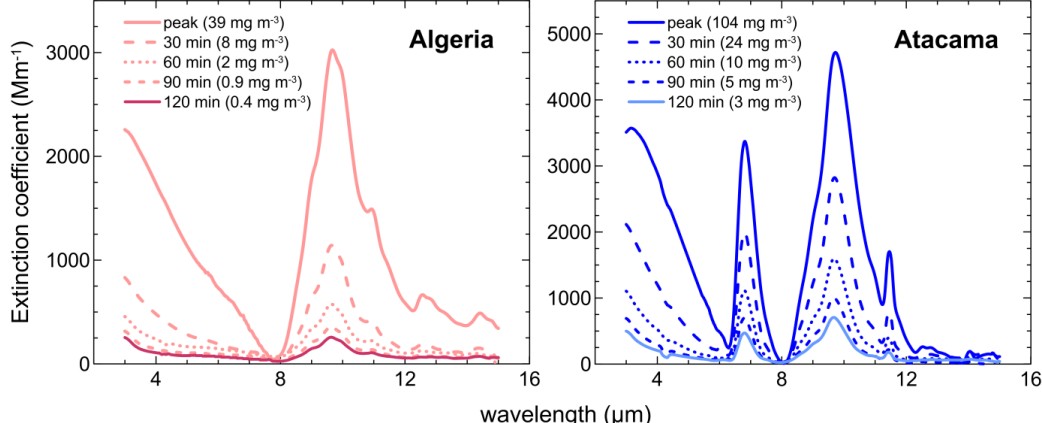




**Figure 10.** Real (n) and imaginary (k) parts of the dust complex refractive index obtained for the nineteen aerosol samples analysed in this study. Data correspond to the time average of the 10-min values obtained between the peak of the injection and 120 min later.

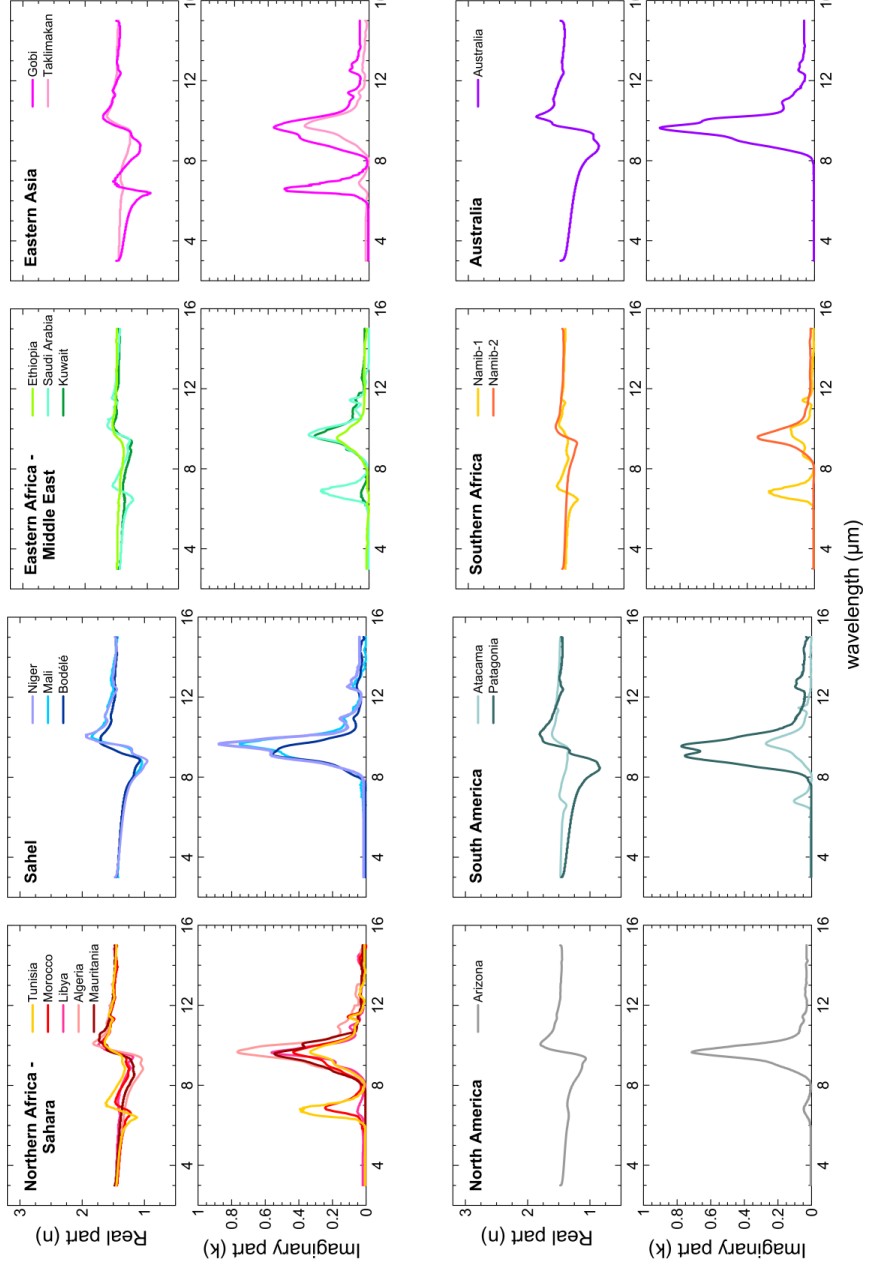





**Figure 11.** Imaginary part of the complex refractive index (k) versus the mineral content (in % mass) for the bands of calcite (7.0 and 11.4 μm), quartz (9.2 μm), and clays (9.6 and 10.9 μm). For the band at 9.6 μm the plot is drawn separately for total clays, and illite and kaolinite species. The linear fits are also reported for each plot.

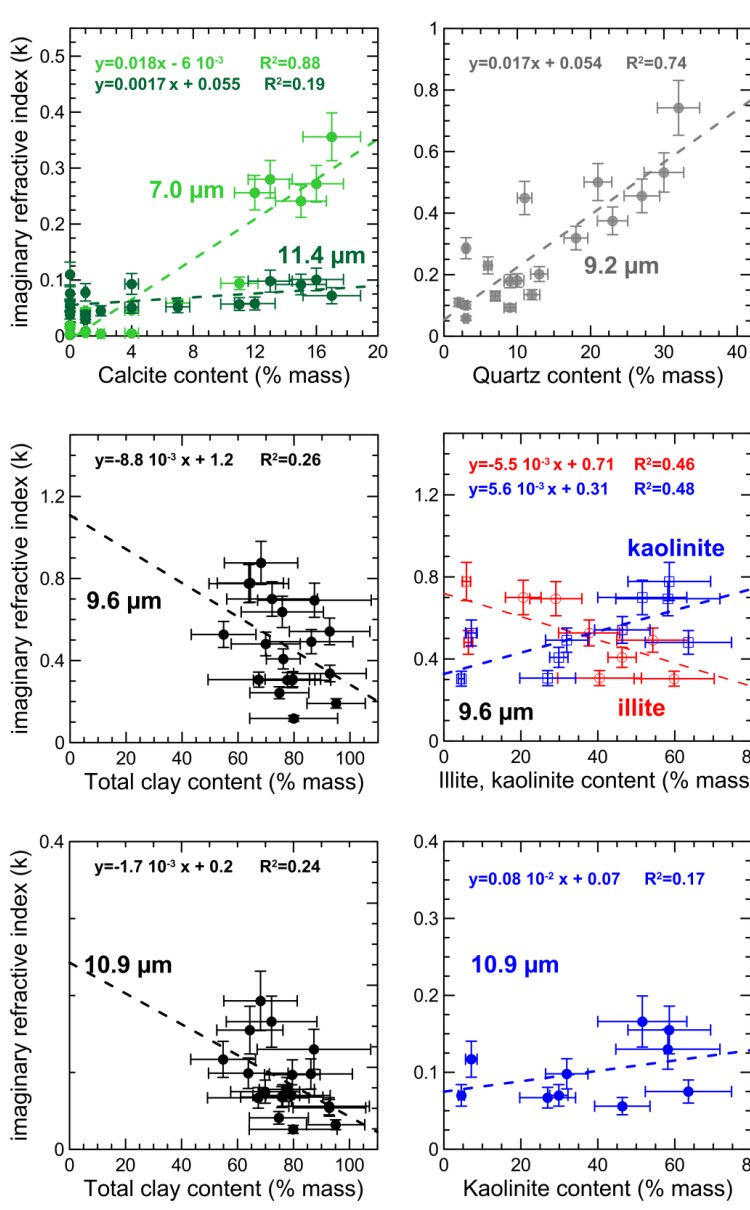



**Figure 12.** Comparison of results obtained in this study with literature values of the dust refractive index in the LW. Literature values are taken from Volz (1972) for rainout dust collected in Germany, Volz (1973) for dust collected at Barbados, Fouquart (1987) for Niger sand, Di Biagio et al (2014a) for dust from Niger and Algeria, and the OPAC database (Hess et al., 1998). The region in gray in the plot indicates the full range of variability obtained in this study, and the dashed line is the mean of n and k obtained for the different aerosol samples. The legend in the top panel identifies the line styles used in the plot for the literature data.

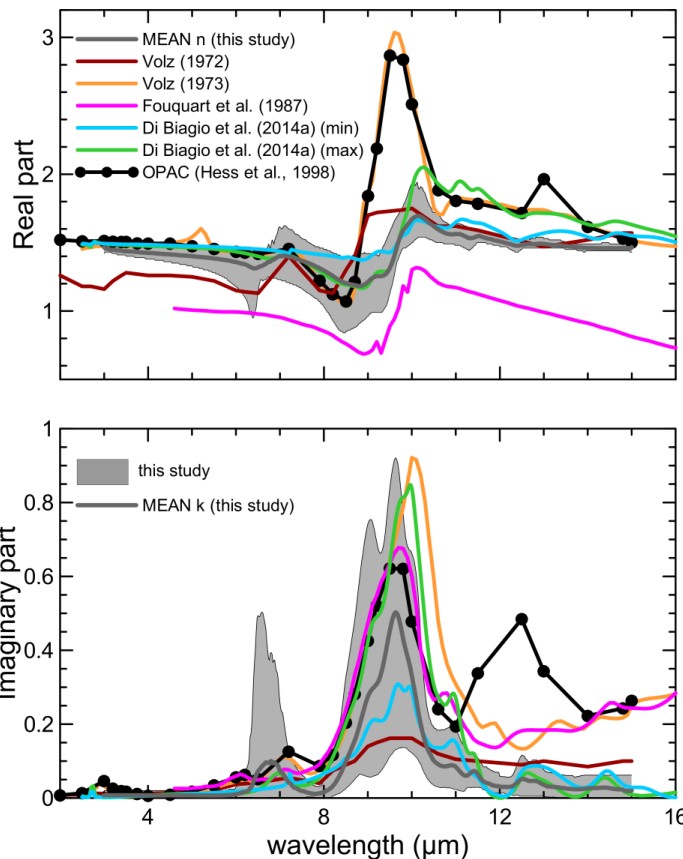



**Figure A1.** Left panel: longwave spectrum of ammonium sulfate measured in CESAM in the 2-15 μm
range. The vibrational modes $v_3(NH_4^+)$ (3230 cm$^{-1}$ or 3.10 μm) and $v_3(SO_2^{-4})$ (1117 cm$^{-1}$ or 8.95
μm) of ammonium sulfate are identified in the plot. Absorption bands attributed to gas-phase water
vapor and $CO_2$ present in the chamber during experiments are also indicated. The rectangle in the plot
indicates the spectral region where the retrieval of the complex refractive index was performed. Right
panel: real and imaginary parts of the refractive index obtained by optical closure. The results are
compared with the ammonium sulfate optical constants from Toon et al. (1976).

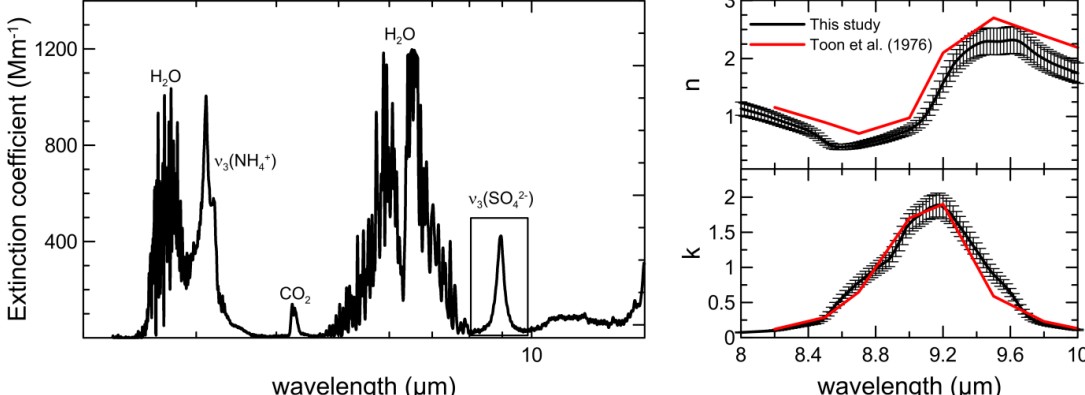
