# Peer review of "Global scale variability of the mineral dust longwave refractive index: 1 a new dataset of in situ measurements for climate modelling and remote sensing 2 3 C. Di Biagio1, P. Formenti1, Y. Balkanski2, L. Caponi1,3, M. Cazaunau1, E. Pangui<s"

_Atmospheric Chemistry and Physics, 2016_

## Referee Comment (RC1) · Anonymous Referee #2 · 8 Nov 2016

**General comments**

This paper presents a new set of desert dust aerosols refractive indices, varying with the source region around the globe. As it shows strong differences with source areas, this new data set is a big step forward in refining the dust models, but also in adjusting dust aerosols retrievals from satellite data. The work is well presented, easy to read, well structured. It contains significant technical information about the experiments undertaken. I am not an expert in laboratory measurements, so it is difficult for me to provide a complete review of this part, especially all the technical choices. However,

everything is clearly explained and understandable, and all seems logical. Uncertainties are discussed together with the results, providing the reader with tools for a full evaluation of the data.

I only have a very short list of comments/questions here under.

**Specific comments**

lines 86-88: could you add a reference here, or some explanation as to why aerosols have those specific effects? The cited paper of Hsu does not mention specificities about the different effects at different altitudes (surface, atmosphere, TOA), and it is not so obvious to me for example why dust LW effect would be to cool the atmosphere (at least within the dust layer where it emits its radiation).

line 102+135 (less important, as e.g. is used) +849: missing Vandenbussche et al, AMT 2013 and one ref from Clarisse et al (many publication about mineral aerosol retrievals); ref to Klüser et al not needed in line 849 indeed, as their retrieval contains a retrieval of mineralogy and is therefore not based on using dust refractive indices

line 105: not so sure it's THE highest uncertainty that comes from there... there is also a huge uncertainty due to altitude or to particle size for example

line 162: why not 79/21? does it change anything?

line 202: here the samples are fully dried (I guess this is to allow generation of the aerosols) while in nature it is obviously not always the case; do the authors have information if / how humidity affects the optical properties of dust aerosols? If it requires a whole additional study it is out of the scope of this paper, obviously.

line 229: why interpolate at such a high spectral resolution? (minerals don't have sharp absorption lines)

line 246: the scattering part is 20% after injection and 10% after 2hours, while in the introduction/abstract it is said that the refractive index does not change with time

line 284: do you have an idea to explain that discrepancy?

line 458: is non-sphericity often used in LW dust retrievals? (not that I know of). Aeronet and Polder are shortwave instruments, where non-sphericity effects are more important

**Technical corrections**

line 416: we combined (7a)-(7B) and ... [the "b" is missing]

line 437: lower thaN

line 456: This assumption could be, however, not fully -> This assumption could, however, not be fully

line 823: there is an unnecessary - at the end of the line

---

## Referee Comment (RC2) · Anonymous Referee #1 · 16 Nov 2016

Review of the manuscript by Di Biagio et al. submitted to ACPD.

Di Biagio et al. have studied the longwave refractive index of mineral dust and how it varies depending on the source region. This is an important topic, especially for climate modelers who need accurate information on the optical properties of mineral dust. The research is done conscientiously using well documented methods. The manuscript is very well written and all the relevant things are discussed. However, some parts of the text should be clarified. My specific comments are given below:

P3L88: What is the net radiative effect (SW+LW) of dust?
P4L115: What processes are you referring to? Oxidation, condensation, accumulation, water uptake?

P8L222-223: Could you give the spectral resolution also in $\mu$m?

P8L229: Could you give the spectral resolution also in cm-1?

P10L291: The usage of Mie theory for non-spherical particles causes errors. For example, the T-matrix method would a more suitable method. Could you estimate how large errors the usage of Mie theory causes?

P12L331: The combining of different size distribution measurements is difficult because different methods produce slightly different results. Did you check how well the SMPS, SkyGrimm and WELAS agreed on the overlapping size ranges and were there large differences? Why did you use the instruments separately for different size ranges? You could have also calculated a combined size distribution for the overlapping size ranges.

P14L394: Again, I'm a bit worried that you use Mie theory for non-spherical particles. It could cause large errors. You should at least discuss the magnitude of the possible uncertainties.

P14L399: What is $\Omega$?

P15L427: Mention here that the uncertainties caused by this choice are discussed in section 3.1.

P16L454: Estimate the magnitude of the uncertainty caused by the use of Mie theory.

P17L483-484: Not sure what this sentence means. Please, clarify.

P17L485: Couldn't you estimate the uncertainty in the refractive index by calculating it using the size distribution and absorption data with maximum uncertainties? So, basically you add the uncertainties to the size and absorption data and see how much they change the calculated refractive index?

P17L488: The uncertainty of 10-20% in the refractive indices sounds surprisingly small when the uncertainty in the number concentration can be as high as 70 %. Why the uncertainty is so small?

P18L499: If the effect of the measured size distribution for the sizes larger than 8 $\mu$m was only 10 % why did you decide to use extrapolated data?

P19L523-525: This is a confusing sentence because you only mention 15 samples. What about the rest of the 19 discussed earlier?

P21L584: Doesn't this complication with the composition reflect also to the representativity of the refractive index results reported?

P21L592: What are the uncertainties here? They would help the comparison.

P21L599: The Eq. 5 has a number distribution. Why are you using surface size distribution here?

P22L623: Why Algeria and Atacama were selected for the comparison with Northern Africa data?

P22L631: I wouldn't say that the particle fractions are comparable for the particles smaller than $\sim$0.4 $\mu$m.

P22L632: This sentence is a bit confusing regarding the size ranges. Please, clarify.

P22L636: Could the extrapolation of the size data have an effect on the difference in deposition? It also started from 8 $\mu$m.

P24L679: Again, why were Algeria and Atacama chosen as examples?

P25L705: You mention standard deviations here. What about the uncertainties? I think they should also be considered. Do the refractive indices within the regions differ more than their uncertainties?

P25L709: Is the variability in the refractive indices larger between the regions than

within the regions? I just interested to know that could the modelers use a single refractive index for a region or do they have to also consider the variability within the regions?

P26L746: Just a comment regarding the linear fits shown in Fig. 11. You don't mention what kind of a fit you used but I hope it wasn't OLS because it is known to cause biases. See the paper by Pitkänen et al. (2016) for more details: http://onlinelibrary.wiley.com/doi/10.1002/2016GL070852/full

P26L758: "short or medium ranges" - What do these mean in km or time in the atmosphere?

P27L765: In this section I would like to see more discussion on the effect of these differences between the reported refractive indices for modelling and remote sensing applications. Are the differences large enough to have a significant effect, for example, on radiative transfer.

P28L814: Source-specific values for the source regions or even within the source regions?

P29L823: This could challenging due to atmospheric absorption. For example, ozone is a strong absorber at 9.6 $\mu$m.

P30L851: Are the regional differences larger than the uncertainties in the LW refractive index?

Table 1: SkyGrimm: Are the particles assumed spherical and if so, what is the uncertainty due to it?

Figure 6: This is a busy figure. What is min and max based on? Sometimes the min is larger than the corresponding max. Shouldn't the CESAM average include only the African and Asian data for a more direct comparison with the previous studies?

Figure 8: The scale for the Northern Africa subplot is different from the others. Same

scale would make comparisons easier.

References: Both Perlwitz papers have the same title.

---

## Author Comment (AC1) · 11 Jan 2017

**Revision of the paper "Global scale variability of the mineral dust longwave refractive index: a new dataset of in situ measurements for climate modelling and remote sensing" by C. Di Biagio et al. (comments in black, answers in red)**

At first, we would like to thank the reviewers for having read the paper and provided valuable comments, which helped to improve the quality of the manuscript. We have taken into consideration all the questions raised by the reviewers, and changed the paper accordingly. The details of our changes are highlighted in the text. The point by point answers to Reviewer #1 and #2 are provided in the following.

During the revision phase, we performed additional measurements of the dust composition and based on these results we performed two modifications compared to the first analysis:

1. Compared to the first version of the paper, the mineralogy of the dust samples was updated by adding the particle iron oxide content. Full details on the retrieval of these data will be provided in another paper (Caponi et al., submitted to ACP). The inclusion of iron oxides data completes the mineralogical composition of dust, but it does not change the results of the paper, given that iron oxides account only for about <4% of the total mass;

2. We also re-evaluated the estimate of the total clay fraction. The clay mass is calculated in this study as the difference between the total mass and the mass of all other minerals detected by XRD (X-Ray Diffraction). We combined two different techniques to estimate the total mass on filters: the mass calculated from the size distribution and the mass estimated from the elemental particle composition obtained from XRF (X-Ray Fluorescence). We found that the mass from the size and the elemental mass of dust were not always in agreement, and so we decided to estimate the clay mass as the mean of the values obtained by the two methods (while in the first version of the paper the clay fraction was estimated only by using the mass from the size distribution). Even if more statistically uncertain, this new estimate of the clay mass is more appropriate, in particular by taking into account the uncertainty in the mass estimation. Table S2 in the Supplementary material reports data on the full dust mineralogy, also reporting the upper and lower limits of the clay content as calculated by assuming the two different mass estimates.

The details of these changes are presented in Section 2.4. All figures, tables, and results affected by changes in mineralogical data have been updated.

*Caponi, L., Formenti, P., Massabó, D., Di Biagio, C., Cazaunau, M., Pangui, E., Chevailler, S., Landrot, G., Fonda, E., Andreae, M. O., B., Kandler, Piketh, S., Saeed, T., Seibert, D., Williams, E., Balkanski, Y., and Doussin, J.-F.: Spectral- and size-resolved mass absorption cross-sections of mineral dust aerosols in the shortwave: a smog chamber study, Atmos. Chem. Phys. Discuss., submitted.*

**Reviewer #1**

Di Biagio et al. have studied the longwave refractive index of mineral dust and how it varies depending on the source region. This is an important topic, especially for climate modelers who need accurate information on the optical properties of mineral dust. The research is done conscientiously using well documented methods. The manuscript is very well written and all the relevant things are discussed. However, some parts of the text should be clarified. My specific comments are given below:

P3L88: What is the net radiative effect (SW+LW) of dust?

To answer this question, the following sentence was added:

"The net effect of dust at TOA is generally a warming over bright surfaces (e.g., deserts) (Yang et al., 2009) and a cooling over dark surfaces (e.g., oceans) (di Sarra et al., 2011)."

P4L115: What processes are you referring to? Oxidation, condensation, accumulation, water uptake?

Processes responsible for particle aging in the atmosphere are now explicitly mentioned. The sentence was re-written as:

"Additional variability is expected to be introduced during transport due to the progressive loss of coarse particles by gravitational settling and chemical processing (particle mixing, heterogeneous reactions, water uptake), which both change the composition of the particles (Pye et al., 1987; Usher et al., 2003)."

P8L222-223: Could you give the spectral resolution also in μm?

The resolution in μm was added in the text.

P8L229: Could you give the spectral resolution also in cm-1?

The resolution in $cm^{-1}$ was added in the text.

P10L291: The usage of Mie theory for non-spherical particles causes errors. For example, the T-matrix method would a more suitable method. Could you estimate how large errors the usage of Mie theory causes?

In this study, we have decided to neglect the non-sphericity of mineral dust to correct optical particle counter diameters for two reasons:

1. first, performing the optical-to-geometrical diameter conversion with the T-matrix theory would require an accurate knowledge of the shape of the particles. As shown by Mishchenko et al. (1997), in fact, the phase function of non-spherical particles is strongly sensitive to the particle aspect ratio (i.e. the ratio of the larger to the shorter dust dimensions). Chou et al. (2008) has shown that the aspect ratio of dust may vary in the wide range 1 to 5, which means

that either the dust shape has to be accurately characterized (which was not the case for our experiments), or the uncertainties due to the fact of using a wrong aspect ratio in the calculations risk to be comparable or even larger than the uncertainties due to the use of Mie theory.

Based on this consideration, by consequence, it is also very hard to estimate the uncertainty induced on the dust size due to the fact of using the Mie theory instead of the T-matrix theory in the calculations.

2. Second, the use of Mie theory was assumed also for sake of comparison with field data published so far (to our knowledge no one uses the T-matrix theory to correct size distribution data). So, the need of validating dust size distributions and particle lifetime against field data requires having comparable datasets and diameter ranges.

*Chou, C., P. Formenti, M. Maille, P. Ausset, G. Helas, M. Harrison, and S. Osborne, Size distribution, shape, and composition of mineral dust aerosols collected during the African Monsoon Multidisciplinary Analysis Special Observation Period 0: Dust and Biomass-Burning Experiment field campaign in Niger, January 2006, J. Geophys. Res., 113, D00C10, doi:10.1029/2008JD009897, 2008.*

*Mishchenko, M. I., L. D. Travis, R. A. Kahn, and R. A. West, Modeling phase functions for dustlike tropospheric aerosols using a shape mixture of randomly oriented polydisperse spheroids, J. Geophys. Res., 102, 16831–16848, 1997.*

P12L331: The combining of different size distribution measurements is difficult because different methods produce slightly different results. Did you check how well the SMPS, SkyGrimm and WELAS agreed on the overlapping size ranges and were there large differences? Why did you use the instruments separately for different size ranges? You could have also calculated a combined size distribution for the overlapping size ranges.

We agree with the reviewer that working with many instruments may be quite complicated due to the different corrections to apply and the various artefacts to take into account. Anyhow, for particles such as mineral dust which present sizes extending over the wide range ~0.1-100 μm, the only approach is to combine instruments based on different principles and operating over different size ranges. As explained in Section 2.3, the different instruments used in chamber experiments (SMPS, WELAS, SkyGrimm) presented also various problems which prevented a full intercomparison, and also the calculation of a combined size in their overlapping intervals. For example, the SkyGrimm had a poor calibration above 1 μm, while the WELAS has a nominally poor counting efficiency below 1 μm; this means that data above/below 1μm for the SkyGrimm/WELAS had to be discarded. So, by consequence, no data were available in an overlapping diameter range for these two instruments. Regarding the SMPS and the SkyGrimm, the SMPS diameter range after density and shape factor correction was 0.01-0.3 μm, while the SkyGrimm measured between 0.29 and 68.2 μm after refractive index correction. So, also in this case in practice there is no overlapping.

Anyhow, to let the reader look at the data from each of the three instruments and have an idea of their agreement, we provided in Figure 3 in the Supplement a plot of size data separately for the different instruments.

P14L394: Again, I'm a bit worried that you use Mie theory for non-spherical particles. It could cause large errors. You should at least discuss the magnitude of the possible uncertainties.

As discussed now more extensively in the text, the use of Mie theory in our retrieval is appropriate given that our measured signal for dust is mostly dominated by absorption (as explained in Sect. 2.2). In this case, the non-sphericity of dust has negligible effects. In fact, as discussed by Kalashnikova and Sokolik (2004), when the size parameter is larger than about 5 (which for example corresponds to 6.3 μm wavelength for 5 μm particle radius, or 9.4 μm for 7.5 μm particle radius), the extinction efficiency of non-spherical particles is systematically higher than spheres due to larger scattering efficiencies, whereas absorption efficiencies are very similar to those of spheres.

To explain this point we added the following text in Section 3.1:

"First, our optical calculations (Eq. (5)) use Mie theory for spherical particles. This is expected to introduce some degrees of uncertainties in simulated LW spectra, especially near the resonant peaks (Legrand et al., 2014). However, as discussed in Kalashnikova and Sokolik (2004), deviations from spherical behavior are mostly due to the scattering component of extinction since irregularly-shaped particles have larger scattering efficiencies than spheres. In contrast, particle absorption is much less sensitive to particle shape. Given that our measured spectra are dominated by absorption, we can therefore reasonably assume that Mie theory is well suited to model our optical data. It also has to be pointed out that at present almost all climate models use Mie theory to calculate dust optical properties. So, with the aim of implementing our retrieved refractive indices in model schemes, it is required that the same optical assumptions are done in both cases, i.e., the optical theory used in models and that used for refractive index retrieval."

P14L399: What is $\Omega$?

$\Omega$ is the generic symbol representing the angular frequency of radiation in the integral expression, with $\omega$ representing the angular frequency of radiation at which the refractive index is known or searched.

P15L427: Mention here that the uncertainties caused by this choice are discussed in section 3.1.

This was added to the text.

P16L454: Estimate the magnitude of the uncertainty caused by the use of Mie theory.

See the answer to one of previous comments.

P17L483-484: Not sure what this sentence means. Please, clarify.

This paragraph was re-written as:

"The uncertainty in the retrieved refractive index was estimated with a sensitivity analysis. Towards this goal, n and k were also obtained by using as input to the retrieval algorithm the measured $\beta_{abs}(\lambda)$ and size distribution ± their estimated uncertainties. The differences between the so obtained n and k and the n and k from the first inversion were estimated. Then, we computed a quadratic combination of these different factors to deduce the uncertainty in n and k."

P17L485: Couldn't you estimate the uncertainty in the refractive index by calculating it using the size distribution and absorption data with maximum uncertainties? So, basically you add the uncertainties to the size and absorption data and see how much they change the calculated refractive index?

Yes, this is what was already done, as explained in Section 3.2. See also the answer to the previous point.

P17L488: The uncertainty of 10-20% in the refractive indices sounds surprisingly small when the uncertainty in the number concentration can be as high as 70 %. Why the uncertainty is so small?

What was not mentioned in the paper is that the uncertainty in the number concentration is size dependent, and values larger than about 30% were measured almost exclusively in the diameter range 0.5-2.0 µm, while lower uncertainties are observed at larger diameters, which is also the size range that dominates the contribution to longwave absorption. This is why the resulting variability in n and k due to size changes is only 10-20%. This point is now more clearly explained in the text:

"The results of the sensitivity study indicated that the measurement uncertainties on $\beta_{abs}(\lambda)$ (±10%) and the size distribution (absolute uncertainty on the number concentration, ±20-70%, with values larger than 30% found at diameters between about 0.5 and 2.0 µm) have an impact of ~10-20% on the retrieval of n and k."

P18L499: If the effect of the measured size distribution for the sizes larger than 8 µm was only 10 % why did you decide to use extrapolated data?

As discussed in Section 2.3.2 particle losses above 8 µm were >95% for the WELAS, so size data were not available above this diameter. Thus we extrapolated data above this value by performing a lognormal fit which reproduced the shape of the measured size between 3 and 8 µm.

Then in Section 3.2 we analyse the impact of the uncertainties in the measured data between 3 and 8 µm on the extrapolated curve and retrieved refractive index. To this aim, we recalculated the lognormal fit by using WELAS data at 3-8 µm ± their estimated uncertainties

and we used these data as input to the inversion algorithm. We found that a change of the extrapolation curve between these limits has an effect <10% on the retrieved refractive index.

This result is quite good, but it does not modify the fact that above 8 μm size data need to be extrapolated.

P19L523-525: This is a confusing sentence because you only mention 15 samples. What about the rest of the 19 discussed earlier?

The discussion in this Section was slightly modified to clarify of the selection procedure of the 19 soils used in the experiments.

P21L584: Doesn't this complication with the composition reflect also to the representativity of the refractive index results reported?

The complication in the comparison with field data and the representativeness of our data are two different issues.

Concerning the comparison with field data, there are two main points:

1. First, it is quite difficult to find a study which analysed a dust sample from exactly the same sources as we considered.

2. Second, field data may be influenced by particle mixing with other aerosol types, or many dust sources may combine together. In this sense, the comparison with chamber data is not so easy to do, unless specific dust episodes are well documented, as it was the case for the Niger case reported by Formenti et al. (2014) to which we compared our results.

These considerations however do not influence the representativeness of our results. Our dust samples were chosen to represent various sources worldwide with very different compositions. In particular, the source regions were selected to represent the whole range of mineralogical variability at the global scale. So, in this sense, even if each dust sample represents a pure and single-source dust aerosol, the combination of data from all of the nineteen samples could be used to represent all possible sources around the globe, as well as their mixing.

P21L592: What are the uncertainties here? They would help the comparison.

The uncertainties on the estimated clay, quartz, and feldspar masses as obtained in this study were added to the main text. It should be noted that given the recalculation of the mineralogical composition, the mineralogy of our Niger sample slightly changed. The new text now is:

"For a case of intense local erosion at Banizoumbou, they showed that the aerosol is composed of 51% (by volume) clays, 41% quartz, and 3% feldspars. Our Niger sample generated from the soil collected at Banizoumbou, is composed of 51% (±5.1%) (by mass)

clays, 37% (±3%) quartz, and 6% (±0.8%) feldspars, in very good agreement with the field observations. ".

P21L599: The Eq. 5 has a number distribution. Why are you using surface size distribution here?

Equation (5) can be expressed in terms of the particle number or surface size distribution, as:

$$\left(\beta_{abs}(\lambda)\right)_{calc} = \sum_{D_g} \frac{\pi D_g^2}{4} Q_{abs}\left(m, \lambda, D_g\right) \left[\frac{dN}{d\log D_g}\right]_{CESAM} d\log D_g$$

$$\left(\beta_{abs}(\lambda)\right)_{calc} = \sum_{D_g} Q_{abs}\left(m, \lambda, D_g\right) \left[\frac{dS}{d\log D_g}\right]_{CESAM} d\log D_g$$

and this since $\dfrac{dS}{d\log D_g} = \dfrac{\pi D_g^2}{4}\left[\dfrac{dN}{d\log D_g}\right]_{CESAM}$ . This point has been specified in the text (Section 3).

This explains why we chosen to plot the surface size distribution of the particles, since that is the quantity that controls their optical behaviour.

P22L623: Why Algeria and Atacama were selected for the comparison with Northern Africa data?

We chose to plot Algeria and Atacama as an example of two cases which:

(i) are issued from different continents and desert areas, one in the Northern and the other one in the Southern Hemisphere; (ii) had different concentration levels in the chamber. These reasons were specified in the text:

"As an example, Fig. 9 shows the temporal evolution of the measured extinction spectrum for the Algeria and Atacama aerosols. These samples were chosen as representative of different geographic areas and different concentration levels in the chamber.".

P22L631: I wouldn't say that the particle fractions are comparable for the particles smaller than ~0.4 μm.

The text was changed as:

"The comparison indicates that the remaining particle fraction observed 30 minutes after the peak of the injection is comparable to that obtained by Ryder et al. (2013b) for particles between ~0.4 and 3 μm for the Algeria case, and ~0.4 and 8 μm for the Atacama case, but that the depletion is much faster for both smaller and larger particles.".

P22L632: This sentence is a bit confusing regarding the size ranges. Please, clarify.

This sentence was changed, as explained in the answer to the previous comment.

P22L636: Could the extrapolation of the size data have an effect on the difference in deposition? It also started from 8 μm.

Data extrapolation should not strongly influence the results for the deposition rate; in fact, as an example, for the Algeria case CESAM data start to deviate from Ryder et al. (2013) at about 3 μm.

P24L679: Again, why were Algeria and Atacama chosen as examples?

See the answer to one of the previous questions on this topic.

P25L705: You mention standard deviations here. What about the uncertainties? I think they should also be considered. Do the refractive indices within the regions differ more than their uncertainties?

Figure 10 was re-plotted also including the uncertainties in n and k (not their standard deviations, but the absolute uncertainty estimated at ~±20%, as suggested by the reviewer). As evident from this new figure, the imaginary part of the refractive index in most cases differs from source to source also within the same regions when uncertainties are taken into account. Conversely, this is not the case for the real part, which agrees within the error for the different sources and also within the same source region. The only exceptions are bands associated with the absorption of specific minerals, such as 7 μm or 9.2 μm affected by calcite and quartz, respectively. This implies that while a constant n can be probably taken for different sources, different values of k should be used both at the global and at the regional scale.

This discussion has been added at the end of Section 5.3, and also the discussion in the Conclusions and the Abstract were modified accordingly. Some minor changes have been also applied throughout the text to take these new issues into account.

P25L709: Is the variability in the refractive indices larger between the regions than within the regions? I just interested to know that could the modelers use a single refractive index for a region or do they have to also consider the variability within the regions?

We would like to thank the reviewer for this comment, which helped to better highlight the implications of our work. As now discussed explicitly in Section 5.3, in the Conclusions, and in the Abstract the extent of the variability for k is of the same order of magnitude both at the global and at the regional scale, so we recommend modellers to take into account the variability of k at both scales.

P26L746: Just a comment regarding the linear fits shown in Fig. 11. You don't mention what kind of a fit you used but I hope it wasn't OLS because it is known to cause biases. See the paper by Pitkänen et al. (2016) for more details: http://onlinelibrary.wiley.com/doi/10.1002/2016GL070852/full.

Linear fits in Figure 11 had been performed with the Ordinary Least Square (OLS) approach, which does not take into account x-uncertainties. In order to take into account the reviewer suggestion, the fits were repeated by using a routine able to take into account both for y and x uncertainties. We used the IDL routine "fitexy.pro" (ref.: "Numerical Recipes" column: Computer in Physics Vol.6 No.3).We mention now explicitly in the paper (caption of Figure 11) the approach used to calculate the linear fit.

Also note that due to the refinement of calculations on the dust mineralogical composition, the points in Figure 11 have slightly changed and by consequence also the results of the fits.

Inspired by the reviewer's previous comments, we also decided to split Figure 11 in 11a and 11b, where in Fig. 11b we plot the real part of the refractive index versus the percent of mineral content for the calcite, quartz and clays bands, to investigate their possible correlation. This was helpful to strengthen the conclusions of our paper concerning the variability of n and k and their possible application in models and remote sensing retrievals.

P26L758: "short or medium ranges" - What do these mean in km or time in the atmosphere?

Based on the comparison with Ryder et al. (2013) in Figure 7, this means about 1-2 days after emission. This is now specified in the text.

P27L765: In this section I would like to see more discussion on the effect of these differences between the reported refractive indices for modelling and remote sensing applications. Are the differences large enough to have a significant effect, for example, on radiative transfer.

In order to follow the reviewer's suggestion, the following discussion was added to Section 6.3:

"On average, the differences between our mean refractive index and the values reported in the literature are large enough to have a significant effect on radiative transfer. For example, at 10 μm the absolute difference between our retrieved mean k and the k by OPAC and Volz (1973) is between 0.15 and 0.6. Highwood et al. (2003) have estimated that a change of about 0.3 in k at 10 μm, which corresponds to half of the difference we have compared to Volz (1973), may result in up to 3 K change in the modelled sky brightness temperature, the quantity measured by infrared remote sensing. To give a comparison, the same order of brightness temperature difference at 10 μm was found between clear sky and dusty conditions for an optical depth of ~1.5 at 0.55 μm. This example illustrates the sensitivity of the brightness temperature to the differences in the imaginary part of the refractive index that we find between our data and those in the literature. Another example, of even more relevance for climate applications, is provided by Di Biagio et al. (2014a), who have shown that a 0.3

variation in k is sufficient to induce up to ~15% of change of the radiative forcing efficiency at 10 μm at the TOA.".

P28L814: Source-specific values for the source regions or even within the source regions?

Also within the source regions, as now stated in the Conclusions. See also the answers to some of the previous comments.

P29L823: This could challenging due to atmospheric absorption. For example, ozone is a strong absorber at 9.6 μm.

The reviewer is correct; however, there are some bands where this could be envisaged, as for example the band at about 7 μm dominated by calcite absorption. Calcite content could allow discriminating between Sahelian and Saharan sources, for example, and within the Sahara also from specific areas where calcite is more abundant than elsewhere. Following the reviewer's comment, we have rewritten the sentence as:

"In addition, the possibility to relate the mass of minerals to the absorption at specific bands, such as for example the calcite band at ~7 μm, implies that the LW extinction spectra measured from space can be used to distinguish between different dust sources."

P30L851: Are the regional differences larger than the uncertainties in the LW refractive index?

In several cases, yes. See also the answers to some of the previous comments.

Table 1: SkyGrimm: Are the particles assumed spherical and if so, what is the uncertainty due to it?

As explained in one of the previous answers, it is quite hard to provide an accurate estimation of this uncertainty without having a detailed characterization of the shape distribution of dust aerosols.

Anyhow, to estimate the uncertainty in the SkyGrimm size due to the use of Mie theory, I performed a calculation based on data of the phase function versus the particle aspect ratio (AR) as reported by Mishchenko et al. (1997) in their Figure 1. I considered the Mishchenko data for the 443 nm wavelength, which were calculated assuming a refractive index of 1.53-0.0085i, and I considered the case of spherical particles and the particles with their highest reported aspect ratio AR=2.4.

I integrated the phase function in the 30°-150° angle range, which is the range of measurements of the SkyGrimm. Then I calculated the ratio of the obtained integrated phase function for the sphere and the AR=2.4. This ratio gives an idea of the change in SkyGrimm geometrical diameter in the two cases.

The results of the calculations indicate less than 1.6% difference in the two cases, which suggests that the deviation in the optical-to-geometrical diameter correction is negligible if Mie theory is used in place of the T-matrix approach.

*Mishchenko, M. I., L. D. Travis, R. A. Kahn, and R. A. West, Modeling phase functions for dustlike tropospheric aerosols using a shape mixture of randomly oriented polydisperse spheroids, J. Geophys. Res., 102, 16831–16848, 1997.*

Figure 6: This is a busy figure. What is min and max based on? Sometimes the min is larger than the corresponding max. Shouldn't the CESAM average include only the African and Asian data for a more direct comparison with the previous studies?

The min and the max are sometimes switched since data are normalised by dividing through the total surface area of dust. To make the Figure easier to read, instead of plotting the min and max separately, we plotted the range of variability for each dataset (AMMA, SAMUM, FENNEC) and also we considered only CESAM data for Northern Africa, as suggested by the reviewer. By consequence, the ACE-ASIA data were eliminated from the plot.

Figure 8: The scale for the Northern Africa subplot is different from the others. Same scale would make comparisons easier.

Figure 8 was re-drawn by using the same y scale for all plots.

References: Both Perlwitz papers have the same title.

The reference was corrected.

**Reviewer #2**

This paper presents a new set of desert dust aerosols refractive indices, varying with the source region around the globe. As it shows strong differences with source areas, this new data set is a big step forward in refining the dust models, but also in adjusting dust aerosols retrievals from satellite data. The work is well presented, easy to read, well structured. It contains significant technical information about the experiments undertaken. I am not an expert in laboratory measurements, so it is difficult for me to provide a complete review of this part, especially all the technical choices. However, everything is clearly explained and understandable, and all seems logical. Uncertainties are discussed together with the results, providing the reader with tools for a full evaluation of the data.

I only have a very short list of comments/questions here under.

Specific comments

lines 86-88: could you add a reference here, or some explanation as to why aerosols have those specific effects? The cited paper of Hsu does not mention specificities about the different effects at different altitudes (surface, atmosphere, TOA), and it is not so obvious to me for example why dust LW effect would be to cool the atmosphere (at least within the dust layer where it emits its radiation).

I re-wrote this part as:

"The SW and LW terms have opposite effects at the surface, Top-of-Atmosphere (TOA), and within the aerosol layer (Hsu et al., 2000; Slingo et al., 2006).,The dust SW effect is to cool the surface and at the TOA, and to warm the dust layer; conversely, the dust LW effect induces a warming of the surface and TOA, and the cooling of the atmospheric dust layer."

I added in the text the reference by Slingo et al. (2006), who discuss more extensively the dust effects at the surface, TOA, and atmosphere

line 102+135 (less important, as e.g. is used) +849: missing Vandenbussche et al, AMT 2013 and one ref from Clarisse et al (many publication about mineral aerosol retrievals); ref to Klüser et al not needed in line 849 indeed, as their retrieval contains a retrieval of mineralogy and is therefore not based on using dust refractive indices.

I added the Vandenbussche et al. (2013) and Clarisse et al. (2013) references in line 102 and 849. Klüser et al was not cited in line 849, instead.

*Clarisse, L., Coheur, P.-F., Prata, F., Hadji-Lazaro, J., Hurtmans, D., and Clerbaux, C.: A unified approach to infrared aerosol remote sensing and type specification, Atmos. Chem. Phys., 13, 2195-2221, doi:10.5194/acp-13-2195-2013, 2013.*

line 105: not so sure it's THE highest uncertainty that comes from there... there is also a huge uncertainty due to altitude or to particle size for example

I re-wrote as: "One of the factors contributing the highest uncertainty is the poor knowledge regarding the dust spectral complex refractive index (m= n-ik)".

line 162: why not 79/21? does it change anything?

We put 80% $N_2$ and 20% $O_2$ in the chamber to reproduce conditions typical of the atmosphere. However, the exact value of the $N_2/O_2$ mixing is not of relevance for our experiments.

line 202: here the samples are fully dried (I guess this is to allow generation of the aerosols) while in nature it is obviously not always the case; do the authors have information if / how humidity affects the optical properties of dust aerosols? If it requires a whole additional study it is out of the scope of this paper, obviously.

In nature, the higher the soil moisture the larger the cohesion of the particles in the soils, and so the lesser the dust emission from the soil itself. A number of studies have investigated this issue (Ravi et al., 2004, 2006; Neuman et al., 2008). So, by drying the soils during our experiments we wanted to ensure generation which could be powerful enough to make dust reach high concentration levels in the chamber, so to magnify the intensity of the measured LW spectrum.

The fact of not drying the soils would cause a less powerful dust generation. Moreover, changes in soil conditions may have an effect on the emitted dust size distribution. The very few studies available on this topic, however, suggest that mostly the fine fraction of dust, which is of minor relevance in the LW, would be affected (Li and Zhang, 2014).

Li, X., and Zhang, H: Soil moisture effects on sand saltation and dust emission observed over the Horqin Sandy Land area in China, J. Meteorol. Res., 28, 444–452, 2014.

Neuman CM, Sanderson S. Humidity control of particle emissions in aeolian systems. J. Geophys Res-Earth Surf. 2008;113:F2. doi: 10.1029/2007JF000780.

Ravi S, D'Odorico P, et al. On the effect of air humidity on soil susceptibility to wind erosion: The case of air-dry soils. Geophys Res Lett. 2004;31:L09501. doi: 10.1029/2004GL019485.

Ravi S, D'Odorico P. A field-scale analysis of the dependence of wind erosion threshold velocity on air humidity. Geophys Res Lett. 2005;32:L21404. doi: 10.1029/2005GL023675.

line 229: why interpolate at such a high spectral resolution? (minerals don't have sharp absorption lines)

FTIR data were acquired between 625 and 5000 cm$^{-1}$. Having 2 cm$^{-1}$ resolution means a resolution varying from 0.05 μm at 625 cm$^{-1}$ (16 μm) to 0.0008 μm at 5000 cm$^{-1}$ (2.0 μm). So basically, by interpolating at 0.02 μm we degraded the spectral resolution. We decided to interpolate at 0.02 μm to have a fixed wavelength step in our data, also maintaining a

relatively high spectral resolution which could be important for applications were high-resolution data are required, such as satellite inversions from IASI.

line 246: the scattering part is 20% after injection and 10% after 2 hours, while in the introduction/abstract it is said that the refractive index does not change with time.

These percentages refer to the fraction of scattering, which contributes to measured extinction at different times after dust injection in the chamber, which in our algorithm (Eq. 2 in the paper) is accounted for as absorption. The refractive index of dust is estimated in our retrieval algorithm by inverting the absorption and using the size distribution of the particles as input to the algorithm. The fact that the refractive index does not change with time when its scattering component does, suggests that the changes in the scattering are due to changes in particle size distribution (absolute concentration and fraction of coarse particles) and not to changes in the particles' composition. This explains why the refractive index does not change.

line 284: do you have an idea to explain that discrepancy?

The only possible explanation that I have concerns the fact that when sampled in the field dust aerosols may have undergone some level of aging, both physical (settling of largest particles) and chemical (possible formation of coatings). The formation of a coating, even if very thin, would strongly modify the particle shape factor. In the chamber, instead, we look at unprocessed particles, and this could possibly justify the observed discrepancy.

line 458: is non-sphericity often used in LW dust retrievals? (not that I know of). Aeronet and Polder are shortwave instruments, where non-sphericity effects are more important.

We agree with the reviewer that this sentence is not of relevance considering the main objective and discussions within the paper. Thus we decided to eliminate it from the text.

Technical corrections

line 416: we combined (7a)-(7B) and ... [the "b" is missing]

The correction was made.

line 437: lower thaN

The correction was made.

line 456: This assumption could be, however, not fully -> This assumption could, however, not be fully

The correction was made.

line 823: there is an unnecessary - at the end of the line

The correction was made.

---

## Author Comment (AC2) · 11 Jan 2017

The comment was uploaded in the form of a supplement:
http://www.atmos-chem-phys-discuss.net/acp-2016-616/acp-2016-616-AC2-supplement.pdf

---

## Author Comment (AC3) · 11 Jan 2017

[revised manuscript text omitted]

[Figure]

[Figure]

**Figure 11a.** Imaginary part of the complex refractive index (k) versus the mineral content (in % mass)
for the bands of calcite (7.0 and 11.4 μm), quartz (9.2 μm), and clays (9.6 and 10.9 μm). For the band
at 9.6 μm the plot is drawn separately for total clays, and illite and kaolinite species. The linear fits are also reported for each plot. Linear fits were performed with the FITEXY.PRO IDL routine taking into account both x- and y-uncertainties ion the data.

[Figure]

[Figure]

**Figure 11b.** Same as Fig. 11a for the real part of the complex refractive index (n).

[Figure]

**Figure 12.** Comparison of results obtained in this study with literature -compiled values of the dust
refractive index in the LW. Literature values are taken from Volz (1972) for rainout dust collected in
Germany, Volz (1973) for dust collected at Barbados, Fouquart (1987) for Niger sand, Di Biagio et al
(2014a) for dust from Niger and Algeria, and the OPAC database (Hess et al., 1998). The region in
gray in the plot indicates the full range of variability obtained in this study, and the dashed line is the
mean of n and k obtained for the different aerosol samples. The legend in the top panel identifies the
line styles used in the plot for the literature data.

[Figure]

**Figure A1.** Left panel: longwave spectrum of ammonium sulfate measured in CESAM in the 2-15 μm range. The vibrational modes $v_3(NH_4^+)$ (3230 cm$^{-1}$ or 3.10 μm) and $v_3(SO_2^{-4})$ (1117 cm$^{-1}$ or 8.95 μm) of ammonium sulfate are identified in the plot. Absorption bands attributed to gas-phase water vapor and $CO_2$ present in the chamber during experiments are also indicated. The rectangle in the plot indicates the spectral region where the retrieval of the complex refractive index was performed. Right panel: real and imaginary parts of the refractive index obtained by optical closure. The results are compared with the ammonium sulfate optical constants from Toon et al. (1976).

[Figure]